# Online Forecasting of Total-Variation-bounded Sequences

**Dheeraj Baby**
Department of Computer Science
UC Santa Barbara
dheeraj@ucsb.edu

**Yu-Xiang Wang**
Department of Computer Science
UC Santa Barbara
yuxiangw@cs.ucsb.edu

## Abstract

We consider the problem of online forecasting of sequences of length $n$ with total-variation at most $C_n$ using observations contaminated by independent $\sigma$-subgaussian noise. We design an $O(n \log n)$-time algorithm that achieves a cumulative square error of $\tilde{O}(n^{1/3} C_n^{2/3} \sigma^{4/3} + C_n^2)$ with high probability. We also prove a lower bound that matches the upper bound in all parameters (up to a $\log(n)$ factor). To the best of our knowledge, this is the first *polynomial-time* algorithm that achieves the optimal $O(n^{1/3})$ rate in forecasting total variation bounded sequences and the first algorithm that *adapts to unknown $C_n$*. Our proof techniques leverage the special localized structure of Haar wavelet basis and the adaptivity to unknown smoothness parameters in the classical wavelet smoothing [Donoho et al., 1998]. We also compare our model to the rich literature of dynamic regret minimization and nonstationary stochastic optimization, where our problem can be treated as a special case. We show that the workhorse in those settings — online gradient descent and its variants with a fixed restarting schedule — are instances of a class of *linear forecasters* that require a suboptimal regret of $\tilde{\Omega}(\sqrt{n})$. This implies that the use of more adaptive algorithms is necessary to obtain the optimal rate.

## 1 Introduction

Nonparametric regression is a fundamental class of problems that has been studied for more than half a century in statistics and machine learning [Nadaraya, 1964, De Boor et al., 1978, Wahba, 1990, Donoho et al., 1998, Mallat, 1999, Scholkopf and Smola, 2001, Rasmussen and Williams, 2006]. It solves the following problem:

- Let $y_i = f(u_i) + \text{Noise}$ for $i = 1, ..., n$. How can we estimate a function $f$ using data points $(u_1, y_1), ..., (u_n, y_n)$ and the knowledge that $f$ belongs to a function class $\mathcal{F}$?

Function class $\mathcal{F}$ typically imposes only weak regularity assumptions on the function $f$ such as boundedness and smoothness, which makes nonparametric regression widely applicable to many real-life applications especially those with unknown physical processes.

A recent and successful class of nonparametric regression technique called trend filtering [Steidl et al., 2006, Kim et al., 2009, Tibshirani, 2014, Wang et al., 2014] was shown to have the property of *local adaptivity* [Mammen and van de Geer, 1997] in both theory and practice. We say a nonparametric regression technique is *locally adaptive* if it can cater to local differences in smoothness, hence allowing more accurate estimation of functions with varying smoothness and abrupt changes. For example, for functions with bounded total variation (when $\mathcal{F}$ is a total variation class), standard nonparametric regression techniques such as kernel smoothing and smoothing splines have a mean square error (MSE) of $O(n^{-1/2})$ while trend filtering has the optimal $O(n^{-2/3})$.

Trend filtering is, however, a batch learning algorithm where one observes the entire dataset ahead of the time and makes inference about the past. This makes it inapplicable to the many time series problems that motivate the study of trend filtering in the first place [Kim et al., 2009]. These include influenza forecasting, inventory planning, economic policy-making, financial market prediction and so on. In particular, it is unclear whether the advantage of trend filtering methods in estimating functions with heterogeneous smoothness (e.g., sharp changes) would carry over to the online forecasting setting. The focus of this work is in developing theory and algorithms for locally adaptive online forecasting which predicts the immediate future value of a function with heterogeneous smoothness using only noisy observations from the past.

## 1.1 Problem Setup

1. Fix action time intervals $1, 2, ..., n$
2. The player declares a forecasting strategy $\mathcal{A}_i : \mathbb{R}^{i-1} \to \mathbb{R}$ for $i = 1, ..., n$.
3. An adversary chooses a sequence $\theta_{1:n} = [\theta_1, \theta_2, \ldots, \theta_n]^T \in \mathbb{R}^n$.
4. For every time point $i = 1, ..., n$:
   (a) We play $x_i = \mathcal{A}_i(y_1, ..., y_{i-1})$.
   (b) We receive a feedback $y_i = \theta_i + Z_i$, where $Z_i$ is a zero-mean, independent subgaussian noise.
5. At the end, the player suffers a cumulative error $\sum_{i=1}^{n} (x_i - \theta_i)^2$.

Figure 1: *Nonparametric online forecasting model. The focus of the proposed work is to design a forecasting strategy that minimizes the expected cumulative square error. Note that the problem depends a lot on the choice of the sequence $\theta_i$. Our primary interest is on sequences with bounded total variation (TV) so that $\sum_{i=2}^{n} |\theta_i - \theta_{i-1}| \le C_n$, but we will also talk about the adaptivity of our method to easier problems such as forecasting Sobolev and Holder functions.*

We propose a model for nonparametric online forecasting as described in Figure 1. This model can be re-framed in the language of the online convex optimization model with three differences.

1. We consider only quadratic loss functions of the form $\ell_t(x) = (x - \theta_t)^2$.
2. The learner receives independent *noisy* gradient feedback, rather than the exact gradient.
3. The criterion of interest is redefined as the *dynamic regret* [Zinkevich, 2003, Besbes et al., 2015]:

$$R_{\text{dynamic}}(\mathcal{A}, \ell_{1:n}) := \mathbb{E}\left[\sum_{t=1}^{n} \ell_t(x_t)\right] - \sum_{t=1}^{n} \inf_{x_t} \ell_t(x_t).$$

The new criterion is called a dynamic regret because we are now comparing to a stronger dynamic baseline that chooses an optimal $x$ in every round. Of course in general, the dynamic regret will be linear in $n$ [Jadbabaie et al., 2015]. To make the problem non-trivial, we restrict our attention to sequences of $\ell_1, ..., \ell_n$ that are *regular*, which makes it possible to design algorithms with *sublinear* dynamic regret. In particular, we borrow ideas from the nonparametric regression literature and consider sequences $[\theta_1, ..., \theta_n]$ that are discretizations of functions in the continuous domain. Regularity assumptions emerge naturally as we consider canonical functions classes such as the Holder class, Sobolev class and Total Variation classes [see, e.g., Tsybakov, 2008, for a review].

## 1.2 Assumptions

We consolidate all the assumptions used in this work and provide necessary justifications for them.

- (A1) The time horizon for the online learner is known to be $n$.
- (A2) The parameter $\sigma^2$ of subgaussian noise in the observations is known.
- (A3) The ground truth denoted by $\theta_{1:n} = [\theta_1, ..., \theta_n]^T$ has its total variation bounded by some positive $C_n$, i.e., we take $\mathcal{F}$ to be the total variation class $\text{TV}(C_n) := \{\theta_{1:n} \in \mathbb{R}^n : \|D\theta_{1:n}\|_1 \le C_n\}$ where $D$ is the discrete difference operator. Here $D\theta_{1:n} = [\theta_2 - \theta_1, \ldots, \theta_n - \theta_{n-1}]^T$.
- (A4) $|\theta_1| \le U$.

The knowledge of $\sigma^2$ in assumption (A2) is primarily used to get the optimal dependence of $\sigma$ in minimax rate. This assumption can be relaxed in practice by using the Median Absolute Deviation estimator as described in Section 7.5 of Johnstone [2017] to estimate $\sigma^2$ robustly. Assumption (A3) features a samples from a large class of functions with spatially inhomogeneous degree of smoothness. The functions residing in this class need not even be continuous. Our goal is to propose a policy that is locally adaptive whose empirical mean squared error converges at the minimax rate for this function class. We stress that we do *not* assume that the learner knows $C_n$. The problem is open and nontrivial even when $C_n$ is known. Assumption (A4) is very mild as it puts restriction only to the first value of the sequence. This assumption controls the inevitable prediction error for the first point in the sequence.

## 1.3 Our Results

The major contributions of this work are summarized below.

- It is known that the minimax MSE for *smoothing* sequences in the TV class is $\tilde{\Omega}(n^{-2/3})$. This implies a lowerbound of $\tilde{\Omega}(n^{1/3})$ for the dynamic regret in our setting. We present a policy ARROWS (**A**daptive **R**estarting **R**ule for **O**nline averaging using **W**avelet **S**hrinkage) with a nearly minimax dynamic regret $\tilde{O}(n^{1/3})$ and a run-time complexity of $O(n \log n)$.

- We show that a class of forecasting strategies — including the popular Online Gradient Descent (OGD) with fixed restarts [Besbes et al., 2015], moving averages (MA) [Box and Jenkins, 1970] — are fundamentally limited by $\tilde{\Omega}(\sqrt{n})$ regret.

- We also provide a more refined lower bound that characterized the dependence of $U, C_n$ and $\sigma$, which certifies the adaptive optimality of ARROWS in all regimes. The bound also reveals a subtle price to pay when we move from the smoothing problem to the forecasting problem, which indicates the separation of the two problems when $C_n/\sigma \gg n^{1/4}$, a regime where the forecasting problem is *strictly* harder (See Figure 3).

- Lastly, we consider forecasting sequences in Sobolev classes and Holder classes and establish that ARROWS can automatically *adapt* to the optimal regret of these *simpler* function classes as well, while OGD and MA cannot, unless we change their tuning parameter (to behave suboptimally on the TV class).

## 2 Related Work

The topic of this paper sits well in between two amazing bodies of literature: nonparametric regression and online learning. Our results therefore contribute to both fields and hopefully will inspire more interplay between the two communities. Throughout this paper when we refer $\tilde{O}(n^{1/3})$ as the optimal regret, we assume the parameters of the problem are such that it is acheivable (see Figure 3).

**Nonparametric regression.** As we mentioned before, our problem — online nonparametric forecasting — is motivated by the idea of using locally adaptive nonparametric regression for time series forecasting [Mammen and van de Geer, 1997, Kim et al., 2009, Tibshirani, 2014]. It is more challenging than standard nonparametric regression because we do not have access to the data in the future. While our proof techniques make use of several components (e.g., universal shrinkage) from the seminal work in wavelet smoothing [Donoho et al., 1990, 1998], the way we use them to construct and analyze our algorithm is new and more generally applicable for converting non-parametric regression methods to forecasting methods.

**Adaptive Online Learning.** Our problem is also connected to a growing literature on adaptive online learning which aims at matching the performance of a stronger time-varying baseline [Zinkevich, 2003, Hall and Willett, 2013, Besbes et al., 2015, Chen et al., 2018b, Jadbabaie et al., 2015, Hazan and Seshadhri, 2007, Daniely et al., 2015, Yang et al., 2016, Zhang et al., 2018a,b, Chen et al., 2018a]. Many of these settings are highly general and we can apply their algorithms directly to our problem, but to the best of our knowledge, none of them achieves the optimal $\tilde{O}(n^{1/3})$ dynamic regret.

In the remainder of this section, we focus our discussion on how to apply the regret bounds in non-stationary stochastic optimization [Besbes et al., 2015, Chen et al., 2018b] to our problem while leaving more elaborate discussion with respect to alternative models (e.g. the constrained comparator

approach [Zinkevich, 2003, Hall and Willett, 2013], adaptive regret [Jadbabaie et al., 2015, Zhang et al., 2018a], competitive ratio [Bansal et al., 2015, Chen et al., 2018a]), as well as the comparison to the classical time series models to Appendix A.

**Regret from Non-Stationary Stochastic Optimization** The problem of non-stationary stochastic optimization is more general than our model because instead of considering only the quadratic functions, $\ell_t(x) = (x - \theta_t)^2$, they work with the more general class of strongly convex functions and general convex functions. They also consider both noisy gradient feedbacks (stochastic first order oracle) and noisy function value feedbacks (stochastic zeroth order oracle).

In particular, Besbes et al. [2015] define a quantity $V_n$ which captures the total amount of "variation" of the functions $\ell_{1:n}$ using $V_n := \sum_{i=1}^{n-1} \|\ell_{i+1} - \ell_i\|_\infty$. [1] Chen et al. [2018b] generalize the notion to $V_n(p,q) := \left( \sum_{i=1}^{n-1} \|\ell_{i+1} - \ell_i\|_p^q \right)^{1/q}$ for any $1 \le p, q \le +\infty$ where $\|\cdot\|_p := (\int |\cdot(x)|^p dx)^{1/p}$ is the standard $L_p$ norm for functions[2]. Table 1 summarizes the known results under the non-stationary stochastic optimization setting.

Table 1: Summary of known minimax dynamic regret in the non-stationary stochastic optimization model. Note that the choice of $q$ does not affect the minimax rate in any way, but the choice of $p$ does. "-" indicates that the no upper or lower bounds are known for that setting.

| Assumptions on $\ell_{1:n}$ | Noisy gradient feedback | | Noisy function value feedback | |
|---|---|---|---|---|
| | $p = +\infty$ | $1 \le p < +\infty$ | $p = +\infty$ | $1 \le p < +\infty$ |
| Convex & Lipschitz | $\Theta(n^{2/3} V_n^{1/3})$ | $O(n^{\frac{2p+d}{3p+d}} V_n(p,q)^{\frac{p}{3p+d}})$ | - | - |
| Strongly convex & Smooth | $\Theta(n^{1/2} V_n^{1/2})$ | $\Theta(n^{\frac{2p+d}{4p+d}} V_n(p,q)^{\frac{2p}{4p+d}})$ | $\Theta(n^{2/3} V_n^{1/3})$ | $\Theta(n^{\frac{4p+d}{6p+d}} V_n(p,q)^{\frac{2p}{6p+d}})$ |

Our assumption on the underlying trend $\theta_{1:n} \in \mathcal{F}$ can be used to construct an upper bound of this quantity of variation $V_n$ or $V_n(p,q)$. As a result, the algorithms in non-stationary stochastic optimization and their dynamic regret bounds in Table 1 will apply to our problem (modulo additional restrictions on bounded domain). However, our preliminary investigation suggests that this direct reduction does *not*, in general, lead to optimal algorithms. We illustrate this observation in the following example.

**Example 1.** Let $\mathcal{F}$ be the set of all bounded sequences in the total variation class $TV(1)$. It can be worked out that $V_n(p,q) = O(1)$ for all $p, q$. Therefore the smallest regret from [Besbes et al., 2015, Chen et al., 2018b] is obtained by taking $p \to +\infty$, which gives us a regret of $O(n^{1/2})$. Note that we expect the optimal regret to be $\tilde{O}(n^{1/3})$ according to the theory of locally adaptive nonparametric regression.

In Example 1, we have demonstrated that one cannot achieve the optimal dynamic regret using known results in non-stationary stochastic optimization. We show in section 3.1 that "Restarting OGD" algorithm has a fundamental lower bound of $\tilde{\Omega}(\sqrt{n})$ on dynamic regret in the TV class.

**Online nonparametric regression.** As we finalize our manuscript, it comes to our attention that our problem of interest in Figure 1 can be cast as a special case of the "online nonparametric regression" problem [Rakhlin and Sridharan, 2014, Gaillard and Gerchinovitz, 2015]. The general result of Rakhlin and Sridharan [2014] implies the *existence* of an algorithm that enjoys a $\tilde{O}(n^{1/3})$ regret for the TV class without explicitly constructing one, which shows that $n^{1/3}$ is the minimax rate when $C_n = O(1)$ (see more details in Appendix A). To the best of our knowledge, our proposed algorithm remains the first *polynomial time* algorithm with $\tilde{O}(n^{1/3})$ regret and our results reveal more precise (optimal) upper and lower bounds on all parameters of the problem (see Section 3.4).

## 3 Main results

We are now ready to present our main results.

## 3.1 Limitations of Linear Forecasters

Restarting OGD as discussed in Example 1, fails to achieve the optimal regret in our setting. A curious question to ask is whether it is the algorithm itself that fails or it is an artifact of a potentially suboptimal regret analysis. To answer this, let's consider the class of linear forecasters — estimators that outputs a fixed linear transformation of the observations $y_{1:n}$. The following preliminary result shows that Restarting OGD is a linear forecaster . By the results of Donoho et al. [1998], linear smoothers are fundamentally limited in their ability to estimate functions with heterogeneous smoothness. Since forecasting is harder than smoothing, this limitation gets directly translated to the setting of linear forecasters.

**Proposition 2.** *Online gradient descent with a fixed restart schedule is a linear forecaster. Therefore, it has a dynamic regret of at least $\tilde{\Omega}(\sqrt{n})$.*

*Proof.* First, observe that the stochastic gradient is of form $2(x_t - y_t)$ where $x_t$ is what the agent played at time $t$ and $y_t$ is the noisy observation $\theta_t +$ Independent noise. By the online gradient descent strategy with the fixed restart interval and an inductive argument, $x_t$ is a linear combination of $y_1, ..., y_{t-1}$ for any $t$. Therefore, the entire vector of predictions $x_{1:t}$ is a fixed linear transformation of $y_{1:t-1}$. The fundamental lower bound for linear smoothers from Donoho et al. [1998] implies that this algorithm will have a regret of at least $\tilde{\Omega}(\sqrt{n})$. $\square$

The proposition implies that we will need fundamentally new *nonlinear* algorithmic components to achieve the optimal $O(n^{1/3})$ regret, if it is achievable at all!

## 3.2 Policy

In this section, we present our policy ARROWS (Adaptive Restarting Rule for Online averaging using Wavelet Shrinkage). The following notations are introduced for describing the algorithm.

- $t_h$ denotes start time of the current bin and $t$ be the current time point.

- $\bar{y}_{t_h:t}$ denotes the average of the $y$ values for time steps indexed from $t_h$ to $t$.

- $pad_0(y_{t_h}, ..., y_t)$ denotes the vector $(y_{t_h} - \bar{y}_{t_h:t}, ..., y_t - \bar{y}_{t_h:t})^T$ zero-padded at the end till its length is a power of 2. *i.e*, a re-centered and padded version of observations.

- $T(x)$ where $x$ is a sequence of values, denotes the element-wise soft thresholding of the sequence with threshold $\sigma\sqrt{\beta \log(n)}$

- H denotes the orthogonal discrete Haar wavelet transform matrix of proper dimensions

- Let $Hx = \alpha = [\alpha_1, \alpha_2, ..., \alpha_k]^T$ where $k$ being a power of 2 is the length of $x$. Then the vector $[\alpha_2, ..., \alpha_k]^T$ can be viewed as a concatenation of $\log_2 k$ contiguous blocks represented by $\alpha[l], l = 0, ..., \log_2(k) - 1$. Each block $\alpha[l]$ at level $l$ contains $2^l$ coefficients.

---

ARROWS: inputs - observed $y$ values, time horizon $n$, std deviation $\sigma$, $\delta \in (0, 1]$, a hyper-parameter $\beta > 24$

1. Initialize $t_h = 1, newBin = 1, y_0 = 0$
2. For $t = 1$ to $n$:
   (a) If $newBin == 1$, predict $x_t^{t_h} = y_{t-1}$, else predict $x_t^{t_h} = \bar{y}_{t_h:t-1}$
   (b) set $newBin = 0$, observe $y_t$ and suffer loss $(x_t^{t_h} - \theta_t)^2$
   (c) Let $\tilde{y} = pad_0(y_{t_h}, ..., y_t)$ and $k$ be the padded length.
   (d) Let $\hat{\alpha}(t_h : t) = T(H\tilde{y})$
   (e) Restart Rule: If $\frac{1}{\sqrt{k}} \sum_{l=0}^{\log_2(k)-1} 2^{l/2} \|\hat{\alpha}(t_h : t)[l]\|_1 > \frac{\sigma}{\sqrt{k}}$ then
       i. set $newBin = 1$
       ii. set $t_h = t + 1$

---

Our policy is the byproduct of following question: How can one lift a batch estimator that is minimax over the TV class to a minimax online algorithm?

Restarting OGD when applied to our setting with squared error losses reduces to partitioning the duration of game into fixed size chunks and outputting online averages. As described in Section 3.1, this leads to suboptimal regret. However, the notion of averaging is still a good idea to keep. If within a time interval, the Total Variation (TV) is adequately small, then outputting sample averages is reasonable for minimizing the cumulative squared error. Once we encounter a bump in the variation, a good strategy is to restart the averaging procedure. Thus we need to adaptively detect intervals with low TV. For accomplishing this, we communicate with an oracle estimator whose output can be used to construct a lowerbound of TV within an interval. The decision to restart online averaging is based on the estimate of TV computed using this oracle. Such a decision rule introduces non-linearity and hence breaks free of the suboptimal world of linear forecasters.

The oracle estimator we consider here is a slightly modified version of the soft thresholding estimator from Donoho [1995]. We capture the high level intuition behind steps (d) and (e) as follows. Computation of Haar coefficients involves smoothing adjacent regions of a signal and taking difference between them. So we can expect to construct a lowerbound of the total variation $\|D\theta_{1:n}\|_1$ from these coeffcients. The extra thresholding step $T(.)$ in (d) is done to denoise the Haar coefficients computed from noisy data. In step (e), a weighted L1 norm of denoised coefficients is used to lowerbound the total variation of the true signal. The multiplicative factors $2^{l/2}$ are introduced to make the lowerbound tighter. We restart online averaging once we detect a large enough variation. The first coefficient $\hat{\alpha}(t_h : t)_1$ is zero due to the re-centering caused by $pad_0$ operation. The hyper-parameter $\beta$ controls the degree to which we shrink the noisy wavelet coefficients. For sufficiently small $\beta$, It is almost equivalent to the universal soft-thresholding of [Donoho, 1995]. The optimal selection of $\beta$ is described in Theorem 3.

We refer to the duration between two consecutive restarts inclusive of the first restart but exclusive of the second as a bin. The policy identifies several bins across time, whose width is adaptively chosen.

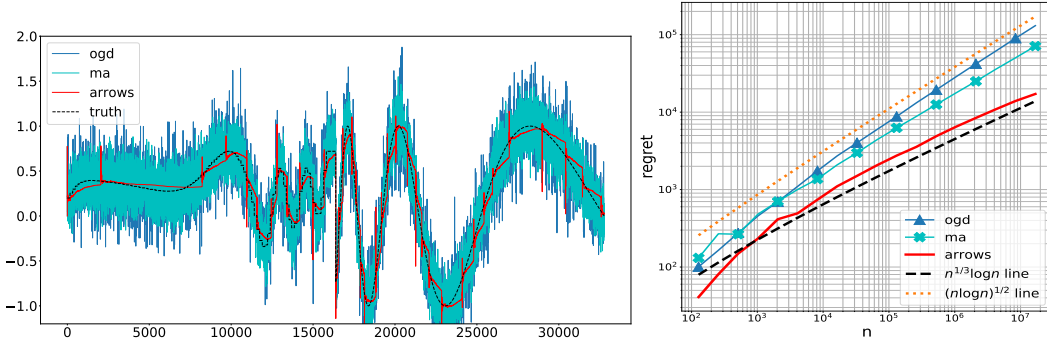

Figure 2: *An illustration of* ARROWS *on a sequence with heterogeneous smoothness. We compare qualitatively (on the left) and quantitatively (on the right) to two popular baselines: (a) restarting online gradient descent [Besbes et al., 2015]; (b) the moving averages [Box and Jenkins, 1970] with optimal parameter choices. As we can see,* ARROWS *achieves the optimal* $\tilde{O}(n^{1/3})$ *regret while the baselines are both suboptimal.*

## 3.3 Dynamic Regret of ARROWS

In this section, we provide bounds for non-stationary regret and run-time of the policy.

**Theorem 3.** *Let the feedback be* $y_t = \theta_t + Z_t$, $t = 1, \ldots, n$ *and* $Z_t$ *be independent,* $\sigma$-*subgaussian random variables. If* $\beta = 24 + \frac{8\log(8/\delta)}{\log(n)}$, *then with probability at least* $1 - \delta$, ARROWS *achieves a dynamic regret of* $\tilde{O}(n^{1/3}\|D\theta_{1:n}\|_1^{2/3}\sigma^{4/3} + |\theta_1|^2 + \|D\theta_{1:n}\|_2^2 + \sigma^2)$ *where* $\tilde{O}$ *hides a logarithmic factor in* $n$ *and* $1/\delta$.

*Proof Sketch.* Our policy is similar in spirit to restarting OGD but with an adaptive restart schedule. The key idea we used is to reduce the dynamic regret of our policy in probability roughly to a sum of squared error of a soft thresholding estimator and number of restarts. This was accomplished by using a Follow The Leader (FTL) reduction. For bounding the squared error part of the sum we modified

the threshold value for the estimator in Donoho [1995] and proved high probability guarantees for the convergence of its empirical mean. To bound the number of times we restart, we first establish a connection between Haar coefficients and total variation. This is intuitive since computation of Haar coefficients can be viewed as smoothing the adjacent regions of a signal and taking their difference. Then we exploit a special condition called "uniform shrinkage" of the soft-thresholding estimator which helps to optimally bound the number of restarts with high probability. □

Theorem 3 provides an upper bound of the minimax dynamic regret for forecasting the TV class.

**Corollary 4.** *Suppose the ground truth $\theta_{1:n} \in TV(C_n)$ and $|\theta_1| \leq U$. Then $\|D\theta_{1:n}\|_1 \leq C_n$. By noting that $\|D\theta_{1:n}\|_2 \leq \|D\theta_{1:n}\|_1$, under the setup in Theorem 3* ARROWS *achieves a dynamic regret of $\tilde{O}(n^{1/3}C_n^{2/3}\sigma^{4/3} + U^2 + C_n^2 + \sigma^2)$ with probability at-least $1 - \delta$.*

**Remark 5** (Adaptivity to unknown parameters.)**.** Observe that ARROWS does not require the knowledge of $C_n$. It adapts optimally to the unknown TV radius $C_n := \|D\theta_{1:n}\|_1$ of the ground truth $\theta_{1:n}$. The adaptivity to $n$ can be achieved by a standard doubling trick. $\sigma$, if unknown, can be robustly estimated from the first few observations by a Median Absolute Deviation estimator (eg. Section 7.5 of Johnstone [2017]), thanks to the sparsity of wavelet coefficients of TV bounded functions.

## 3.4 A lower bound on the minimax regret

We now give a matching lower bound of the expected regret, which establishes that ARROWS is adaptively minimax.

**Proposition 6.** *Assume $\min\{U, C_n\} > 2\pi\sigma$ and $n > 3$, there is a universal constant c such that*

$$\inf_{x_{1:n}} \sup_{\theta_{1:n} \in \text{TV}(C_n)} \mathbb{E}\left[\sum_{t=1}^{n}(x_t(y_{1:t-1}) - \theta_t)^2\right] \geq c(U^2 + C_n^2 + \sigma^2 \log n + n^{1/3}C_n^{2/3}\sigma^{4/3}).$$

The proof is deferred to the Appendix I. The result shows that our result in Theorem 3 is optimal up to a logarithmic term in $n$ and $1/\delta$ for almost all regimes (modulo trivial cases of extremely small $\min\{U, C_n\}/\sigma$ and $n$)[3].

**Remark 7** (The price of forecasting)**.** The result also shows that *forecasting is strictly harder than smoothing*. Observe that a term with $C_n^2$ is required even if $\sigma = 0$, whereas in the case of a one-step look-ahead oracle (or the smoothing algorithm that sees all $n$ observations) does not have this term. This implies that the total amount of variation that *any* algorithm can handle while producing a sublinear regret has dropped from $C_n = o(n)$ to $C_n = o(\sqrt{n})$. Moreover, the regime where the $n^{1/3}C_n^{2/3}\sigma^{4/3}$ term is meaningful only when $C_n = o(n^{1/4})$. For the region where $\sigma n^{1/4} \ll C_n \ll \sigma n^{1/2}$, the minimax regret is essentially proportional to $C_n^2$. This is illustrated in Figure 3.

We note that in much of the online learning literature, it is conventional to consider a slightly more restrictive setting with bounded domain, which could reduce the minimax regret. The following remark summarizes a variant of our results in this setting.

**Remark 8** (Minimax regret in bounded domain)**.** If we consider predicting sequences from a subset of the $TV(C_n)$ ball having an extra boundedness condition $|\theta_i| \leq B$ for $i = 1 \ldots n$, it can be shown that (see Appendix I) minimax regret is $\tilde{\Omega}\left(\min\{nB^2, n\sigma^2, n^{1/3}C_n^{2/3}\sigma^{4/3}\} + B^2 + \min\{nB^2, BC_n\} + \sigma^2\right)$. In particular, forecasting is still strictly harder than smoothing due to the $\min\{nB^2, BC_n\}$ term in the bound. The discussion in Appendix I, shows a way of using ARROWS whose regret can match this lower bound.

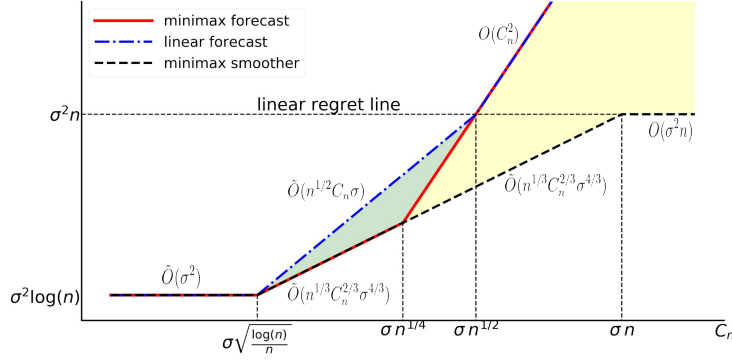

Figure 3: *An illustration of the minimax (dynamic) regret of forecasters and smoothers as a function of $C_n$. The non-trivial regime for forecasting is when $C_n$ lies between $\sigma\sqrt{\frac{\log(n)}{n}}$ and $\sigma\,n^{1/4}$ where forecasting is just as hard as smoothing. When $C_n > \sigma\,n^{1/4}$, forecasting is harder than smoothing. The yellow region indicates the extra loss incurred by any minimax forecaster. The green region marks the extra loss incurred by a linear forecaster compared to minimax forecasting strategy. The figure demonstrates that linear forecasters are sub-optimal even in the non-trivial regime. When $C_n > \sigma\,n^{1/2}$, it is impossible to design a forecasting strategy with sub-linear regret. For $C_n > \sigma\,n$, identity function is optimal estimator for smoothing and when when $C_n < \sigma\sqrt{\frac{\log(n)}{n}}$, online averaging is optimal for both problems.*

## 3.5 The adaptivity of ARROWS to Sobolev and Holder classes

It turns out that ARROWS is also adaptively optimal in forecasting sequences in the discrete Sobolev classes and the discrete Holder classes, which are defined as

$$\mathcal{S}(C'_n) = \{\theta_{1:n} : \|D\theta_{1:n}\|_2 \le C'_n\}, \qquad \mathcal{H}(B'_n) = \{\theta_{1:n} : \|D\theta_{1:n}\|_\infty \le B'_n\}.$$

These classes feature sequences that are more spatially homogeneous than those in the TV class. The minimax cumulative error of nonparametric estimation in the discrete Sobolev class is $\Theta(n^{2/3}[C'_n]^{2/3}\sigma^{4/3})$ [see e.g., Sadhanala et al., 2016, Theorem 5 and 6].

**Corollary 9.** *Let the feedback be $y_t = \theta_t + Z_t$ where $Z_t$ is an independent, $\sigma$-subgaussian random variable. Let $\theta_{1:n} \in \mathcal{S}(C'_n)$ and $|\theta_1| \le U$. If $\beta = 24 + \frac{8\log(8/\delta)}{\log(n)}$, then with probability at least $1 - \delta$, ARROWS achieves a dynamic regret of $\tilde{O}(n^{2/3}[C'_n]^{2/3}\sigma^{4/3} + U^2 + [C'_n]^2 + \sigma^2)$ where $\tilde{O}$ hides a logarithmic factor in $n$ and $1/\delta$.*

Thus despite the fact that ARROWS is designed for total variation class, it adapts to the optimal rates of forecasting sequences that are spatially regular. To gain some intuition, let's minimally expand the Sobolev ball to a TV ball of radius $C_n = \sqrt{n}C'_n$. The chosen scaling of $C_n$ activates the embedding $\mathcal{S}(C'_n) \subset TV(C_n)$ (see the illustration in Table 2) with both classes having same minimax rate in the batch setting. This implies that dynamic regret of ARROWS is simultaneously minimax optimal over $\mathcal{S}(C'_n)$ and $TV(C_n)$ wrt the term containing $n$. It can be shown that ARROWS is optimal wrt to the additive $[C'_n]^2, U^2, \sigma^2$ terms as well. Minimaxity in Sobolev class implies minimaxity in Holder class since it is known that a Holder ball is sandwiched between two Sobolev balls having the same minimax rate [see e.g., Tibshirani, 2015]. A proof of the Corollary and related experiments are presented in Appendix F and J.

## 3.6 Fast computation

Last but not least, we remark that there is a fast implementation of ARROWS that reduces the overall time-complexity for $n$ step from $O(n^2)$ to $O(n \log n)$.

**Proposition 10.** *The run time of ARROWS is $O(n \log(n))$, where $n$ is the time horizon.*

The proof exploits the sequential structure of our policy and sparsity in wavelet transforms, which allows us to have $O(\log n)$ incremental updates in all but $O(\log n)$ steps. See Appendix G for details.

Table 2: *Minimax rates for cumulative error $\sum_{i=1}^{n}(\hat{\theta}_i - \theta)^2$ in various settings and policies that achieve those rates. ARROWS is adaptively minimax across all of the described function classes while linear forecasters fail to perform optimally over the TV class. For simplicity, we assume $U$ is small and hide a $\log n$ factors in all the forecasting rates.*

| Class | | Minimax rate for Forecasting | Minimax rate for Smoothing | Minimax rate for Linear Forecasting |
|---|---|---|---|---|
| TV | $\|D\theta_{1:n}\|_1 \le C_n$ | $n^{1/3}C_n^{2/3}\sigma^{4/3}+C_n^2+\sigma^2$ | $n^{1/3}C_n^{2/3}\sigma^{4/3}+\sigma^2$ | $n^{1/2}C_n\sigma+C_n^2+\sigma^2$ |
| Sobolev | $\|D\theta_{1:n}\|_2 \le C_n'$ | $n^{2/3}[C_n']^{2/3}\sigma^{4/3}+[C_n']^2+\sigma^2$ | $n^{2/3}[C_n']^{2/3}\sigma^{4/3}+\sigma^2$ | $n^{2/3}[C_n']^{2/3}\sigma^{4/3}+[C_n']^2+\sigma^2$ |
| Holder | $\|D\theta_{1:n}\|_\infty \le L_n$ | $nL_n^{2/3}\sigma^{4/3}+nL_n^2+\sigma^2$ | $nL_n^{2/3}\sigma^{4/3}+\sigma^2$ | $nL_n^{2/3}\sigma^{4/3}+nL_n^2+\sigma^2$ |
| Minimax Algorithm | | ARROWS | Wavelet Smoothing Trend Filtering | Restarting OGD Moving Averages |

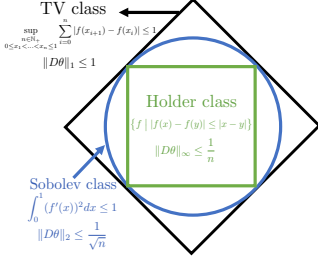

TV class
$\sup_{\substack{u \in \mathbb{R}_n \\ 0 \le x_1 < \cdots < x_n \le 1}} \sum_{i=0}^{n} |f(x_{i+1}) - f(x_i)| \le 1$
$\|D\theta\|_1 \le 1$

Holder class
$\{f \mid |f(x) - f(y)| \le |x - y|\}$
$\|D\theta\|_\infty \le \frac{1}{n}$

Sobolev class
$\int_0^1 (f'(x))^2 dx \le 1$
$\|D\theta\|_2 \le \frac{1}{\sqrt{n}}$

| Canonical Scaling[a] | | Forecasting | Smoothing | Linear Forecasting |
|---|---|---|---|---|
| TV | $C_n \asymp 1$ | $n^{1/3}$ | $n^{1/3}$ | $n^{1/2}$ |
| Sobolev | $C_n' \asymp 1/\sqrt{n}$ | $n^{1/3}$ | $n^{1/3}$ | $n^{1/3}$ |
| Holder | $L_n \asymp 1/n$ | $n^{1/3}$ | $n^{1/3}$ | $n^{1/3}$ |

[a]The "canonical scaling" are obtained by discretizing functions in canonical function classes. Under the canonical scaling, Holder class $\subset$ Sobolev class $\subset$ TV class, as shown in the figure on the left. ARROWS is optimal for the Sobolev and Holder classes inscribed in the TV class. MA and Restarting OGD on the other hand require different parameters and prior knowledge of variational budget (i.e $C_n$ or $C_n'$) to achieve the minimax linear rates for the TV class and the Sobolev/Holder class.

## 3.7 Experimental Results

To empirically validate our results, we conducted a number of numerical simulations that compares the regret of ARROWS, (Restarting) OGD and MA. Figure 2 shows the results on a function with heterogeneous smoothness (see the exact details and more experiments in Appendix B) with the hyper-parameters selected according to their theoretical optimal choice for the TV class (See Theorem 11, 12 for OGD and MA in Appendix C). The left panel illustrates that ARROWS is locally adaptive to heterogeneous smoothness of the ground truth. Red peaks in the figure signifies restarts. During the initial and final duration, the signal varies smoothly and ARROWS chooses a larger window size for online averaging. In the middle, signal varies rather abruptly. Consequently ARROWS chooses a smaller window size. On the other hand, the linear smoothers OGD and MA use a constant width and cannot adapt to the different regions of the space. This differences are also reflected in the quantitative evaluation on the right, which clearly shows that OGD and MA has a suboptimal $\tilde{O}(\sqrt{n})$ regret while ARROWS attains the $\tilde{O}(n^{1/3})$ minimax regret!

## 4 Concluding Discussion

In this paper, we studied the problem of online nonparametric forecasting of bounded variation sequences. We proposed a new forecasting policy ARROWS and proved that it achieves a cumulative square error (or dynamic regret) of $\tilde{O}(n^{1/3}C_n^{2/3}\sigma^{4/3}+\sigma^2+U^2+C_n^2)$ with total runtime of $O(n \log n)$. We also derived a lower bound for forecasting sequences with bounded total variation which matches the upper bound up to a logarithmic term which certifies the optimality of ARROWS in all parameters. Through connection to linear estimation theory, we assert that no linear forecaster can achieve the optimal rate. ARROWS is highly adaptive and has essentially no tuning parameters. We show that it is adaptively minimax (up to a logarithmic factor) simultaneously for all discrete TV classes, Sobolev classes and Holder classes with unknown radius. Future directions include generalizing to higher order TV class and other convex loss functions.

## Acknowledgement

DB and YW were supported by a start-up grant from UCSB CS department and a gift from Amazon Web Services. The authors thank Yining Wang for a preliminary discussion that inspires the work, and Akshay Krishnamurthy and Ryan Tibshirani for helpful comments to an earlier version of the paper.

## Footnotes

[1] The $V_n$ definition in [Besbes et al., 2015] for strongly convex functions are defined a bit differently, the $\|\cdot\|_\infty$ is taken over the convex hull of minimizers. This creates some subtle confusions regarding our results which we explain in details in Appendix I.

[2] We define $V_n(p,q)$ to be a factor of $n^{-1/q}$ times bigger than the original scaling presented in [Chen et al., 2018b] so the results become comparable to that of [Besbes et al., 2015].

[3]When both $U$ and $C_n$ are moderately small relative to $\sigma$, the lower bound will depend on $\sigma$ a little differently because the estimation error goes to 0 faster than $1/\sqrt{n}$. We know the minimax risk exactly for that case as well but it is somewhat messy [see e.g. Wasserman, 2006]. When they are both much smaller than $\sigma$, e.g., when $\min\{U, C_n\} \leq \sigma/\sqrt{n}$, then outputting 0 when we do not have enough information will be better than doing online averages.

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
