[Supplementary Material]

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

# A Discussion on other related works

**Regret from Adaptive Optimistic Mirror Descent.** In Jadbabaie et al. [2015], the authors propose Adaptive Optimistic Mirror Descent (AOMD) algorithm that minimizes the dynamic regret against a comparator sequence $\{u_t\}_{t=1}^n$. Their learning framework is the full information setting where learner predict $x_t \in \mathcal{X}$ for a convex set $\mathcal{X} \subseteq R^d$. Then a loss function $f_t(x)$ is revealed to the learner. To capture the regularity of the comparator, they define a quantity $C_n(u_1, u_2, ..., u_n) := \sum_{t=1}^n \|u_t - u_{t-1}\|$. They capture the regularity of loss functions by incorporating some external knowledge about their gradients via a predictable sequence $\{M_t\}_{t=1}^n$. They define: $D_n := \sum_{t=1}^n \|\nabla f_t(x_t) - M_t\|_*^2$. Finally to account for the temporal variability of $f_t$, they introduce $V_n$ as discussed earlier. The final regret bound is expressed in terms of these three quantities. However, their algorithm is adaptive and requires no prior knowledge about them.

We note that our problem can be reduced to their framework if one considers loss functions $f_t(x) = (x - y_t)^2$. Then the expected dynamic regret against the comparator sequence $\{\theta_t\}_{t=1}^n$ is given by

$$\sum_{t=1}^n E[(x - y_t)^2 - (\theta_t - y_t)^2] = E[\sum_{t=1}^n (x - \theta_t)^2], \tag{1}$$

where the expectation at right hand side is over the randomness of forecasting strategy. Hence we observe that their regret bound can be directly applied to bound the dynamic regret of our problem. It can be shown that (see Appendix H) given a fixed total variation bound $C_n = O(1)$, then $V_n$ and $D_n$ can be proved to be $O(n)$ with high probability. Plugging this into their regret bound yields an $\tilde{O}(\sqrt{n})$ rate in probability. However, it is unclear that whether AOMD is fundamentally limited by this rate or is there a potential suboptimality in their analysis of regret on our particular problem.

**Other Dynamic Regret minimizing policies.** [Yang et al., 2016] defines a path variation budget that is equivalent to our $C_n$ to characterize the sequence of convex loss functions. However, under the noisy gradient feedback structure, they use a version of restarting OGD to get $C^{1/2}n^{1/2}$ regret rate. This is very similar to the policy in [Besbes et al., 2015]. Since OGD is a linear forecaster, it is sub-optimal for predicting bounded variation sequences under the squared error metric.

In [Koolen et al., 2015], they consider minimizing the dynamic regret wrt to a comparator class that obeys $\|D\theta_{1:n}\|_2 \le C'_n$. This is basically the discrete Sobolev class. As shown in appendix E, our policy is minimax for forecasting such sequences as well when the observed values are noisy versions of the ground truth. However it should be noted that [Koolen et al., 2015] does not have this distributional assumption on the observations.

[Chen et al., 2018a] considers the Smoothed Online Convex Optimization framework where they simultaneously minimize the hitting loss $f_t$ and a switching cost. They provide dynamic regret bounds on this composite cost in the setting that $f_t$ is known to the learner before making the prediction. If we consider $f_t(x) = (x - y_t)^2$, then the baseline they compare against reduces to the offline Trend Filtering (TF) estimate when $\sum_{i=2}^n |x_t - x_{t-1}| \le L = C_n$. Then Theorem 10 of [Chen et al., 2018a] states that the cumulative composite cost incurred by their proposed policy differs from that of the TF estimate by a term that is $O(\sqrt{nC_n})$. However, this doesn't translate to a meaningful regret bound in our setting.

[Hall and Willett, 2013] proposes the Dynamic Mirror Descent (DMD) algorithm that make use of a family of dynamical models for making prediction at each time step. They achieve a dynamic regret bound of $O(\sqrt{n}(1 + V_{\phi_t}(\boldsymbol{\theta}_T)))$ where the second term measures the quality of the dynamical models in predicting ground truth.

**Comparison to Online Isotonic Regression.** [Kotłowski et al., 2016] considers the dynamic regret minimization,

$$\sum_{t=1}^n (x_t - y_t)^2 - \min_{(\theta_1, ..., \theta_n)} \sum_{t=1}^n (\theta_t - y_t)^2,$$

where $y_t \le B$ is a label revealed by the environment, $x_t \le B$ is the prediction of the learner, and the comparator sequence should obey $0 \le \theta_1 \le ... \le \theta_n \le B$. Here $B$ is a fixed positive number. Note that their setting and our framework are mutually reducible to each other in terms of regret guarantees

via 1. They propose a minimax policy that achieves a dynamic regret of $\tilde{O}(n^{1/3})$ which translates to an $\tilde{O}(n^{1/3})$ in probability in our setting under the isotonic ground truth restriction.

We note that the isotonic comparator sequence belong to a TV class of variational budget $C_n = B$. By using an argument similar to that in appendix H which involves converting to deterministic noise setting and conditioning on a high probability event, it can be shown that our policy is out of the box minimax with high probability in isotonic framework when observations are noisy versions of an isotonic sequence.

**Comparison to Online Non-Parametric regression methods.** We note that our problem falls into the more general framework of online non-parametric regression setting studied in [Rakhlin and Sridharan, 2015]. We can reduce our dynamic regret minimization to their framework by using a similar argument as above through (1). Since the bounded TV class is sandwiched between Besov spaces $B_{1,q}^1$ for the range $1 \le q \le \infty$, the discussion in section 5.8 of [Rakhlin and Sridharan, 2015] establishes that minimax growth w.r.t $n$ as $O(n^{1/3})$ in the online setting for TV class. Thus our bounds, modulo logarithmic factor, matches with theirs though we give the precise dependence on $C_n$ and $\sigma$ as well. It is worthwhile to point out that while the bound in [Rakhlin and Sridharan, 2015] is non-constructive, we achieve the same bound via an efficient algorithm.

[Gaillard and Gerchinovitz, 2015] proposes a minimax policy wrt to comaparator functions that are Holder smooth. In particular, for the Holder class $H_1$ that satisfy $|f(x) - f(y)| \le \lambda|x - y|$, their algorithm yields a regret of $\tilde{O}(n^{1/3})$. It is known ([Tibshirani, 2015]) that $H_1$ is sandwiched between two Sobolev balls having the same minimax rate in the iid batch statistical learning setting. Since our policy is optimal for Sobolev space (appendix F), it is also optimal over Holder ball $H_1$ when the observations are noisy versions of a Holder smooth functions. Though the framework of [Gaillard and Gerchinovitz, 2015] doesn't impose this distributional assumption. The runtime of their policy for $H_1$ class is $O(n^{7/3} \log n)$. It should be noted that Sobolev and Holder classes are arguably easier to tackle than the TV class since both of them can be embedded inside a TV class.

**Strongly Adaptive Regret.** Daniely et al. [2015] introduced the notion of Strongly Adaptive (SA) regret where the online learner is guaranteed to have low static regret for any interval within the duration of the game. They also propose a meta algorithm which can convert an algorithm of good static regret to one with good SA regret. However low static regret for any interval doesn't help in our setting because in each interval we are competing with a stronger dynamic adversary. A notion of SA dynamic regret would an interesting topic to explore.

For minimizing dynamic regret, Zhang et al. [2018b] proposed a meta policy that uses an algorithm with good SA regret as its subroutine. Hence we can use their framework with squared error loss functions as discussed above. They show that OGD has an SA regret of $O(\log(n))$ for strongly convex loss functions. Using OGD as the subroutine and applying corollary 7 of their paper yields a bound $\tilde{O}(n)$. By a similar argument one gets the same linear regret rate when online newton step is used as the subroutine. However, we should note that their algorithm works without the knowledge of radius of the TV ball $C_n$.

**Classical time series forecasting models.** Finally, we note that our work is complementary to much of the classical work in time-series forecasting (e.g., Box-Jenkins method/ARIMA Box and Jenkins [1970], Hidden Markov Models [Baum and Petrie, 1966]). These methods aim at using dynamical systems to capture the recurrent patterns under a stationary stochastic process, while we focus on harnessing the nonstationarity. Our work is closer to the "trend smoothing" literature (e.g., the celebrated Hodrick-Prescott filter [Hodrick and Prescott, 1997], trend filtering [Kim et al., 2009, Tibshirani, 2014, Hutter and Rigollet, 2016]).

# B   Additional Experiments

The function that we generated in Figure 2 is a hybrid function which in the first half is a "discretized cubic spline" with more knots closely placed towards the end. In the second half it is a Doppler function $f(t) = \sin\left(\frac{2\pi(1+\epsilon)}{t/n+0.38}\right)$ with $n$ being the time horizon. We observe noisy data $y_i =$

Figure 4: *An illustration of* ARROWS *on a linear trend which has homogeneous smoothness*

Figure 5: *An illustration of* ARROWS *on a step trend with abrupt inhomogeneity.*

$f(i/n) + z_i$, $i = 1, ..., n$ and $z_i$ are iid normal variables with $\sigma = 1$. The value of $C_n$ for $n > 60K$ is around 17. Hence for all $n > 83521$, we are under the $n^{1/3}$ regime of $\sigma\sqrt{\log(n)/n} < C_n < \sigma n^{1/4}$.

The window size for moving averages and partition width of OGD were tuned optimally for the TV class (see Appendix C for details). Figure 2 depicts the estimated signals and dynamic regret averaged across 5 runs in a log log plot. The left panel illustrates that ARROWS is locally adaptive to heterogeneous smoothness of the ground truth. Red peaks in the figure signifies restarts. During the initial and final duration, the signal varies smoothly and ARROWS chooses a larger window size for online averaging. In the middle, signal varies rather abruptly. Consequently ARROWS chooses a smaller window size. On the other hand, the linear smoothers OGD and MA attains a suboptimal $\tilde{O}(\sqrt{n})$ regret.

In Figure 4 and 5 we plot the estimates and log-log regret for two more functions: A linear function that is homogeneously smooth and less challenging and a step function which has an abrupt discontinuity making it more inhomogeneous than linear but have lesser inhomogeneity w.r.t hybrid signal considered in 3.7. Both OGD and MA were optimally tuned for the TV class as in Appendix C.

The red peaks corresponds to restarts by ARROWS. For linear functions we can see that ARROWS chooses inter-restart duration/bin-widths that are constant throughout. This is expected as a linear trend is spatially homogeneous. For the step function, we see that ARROWS restart only once since the start. Further, notice that it quickly restarts once the bump is hit. For both of these functions, necessary scaling is done so that we are in the $n^{1/3}$ regime quite early.

# C Upper bounds of linear forecasters

In this section we compute the optimal batch size for Restarting OGD and optimal window size for moving averages to yield the $\tilde{O}(\sqrt{n})$ regret rate.

**Theorem 11.** *Let the feedback be $y_t = \theta_t + Z_t$ where $Z_t$ is an independent, $\sigma$-subgaussian random variable. Let $\theta_{1:n} \in \mathrm{TV}(C_n)$. Restarting OGD with batch size of $\sqrt{n \log n} \frac{\sigma}{C_n}$ achieves an expected dynamic regret of $\tilde{O}(U^2 + C_n^2 + \sigma C_n \sqrt{n})$.*

*Proof.* Note that in our setting with squared error losses $f_t(x) = (x - \theta_t)^2$, the update rule of restarting OGD reduces to computing online averages. Thus OGD essentially divides the time horizon $n$ into fixed size batches and output online averages within each batch. Our objective here is to compute the optimal batch size that minimizes the dynamic regret.

We will bound the expected regret. Let $x_t$ be the estimate of OGD at time $t$. Let batches be numbered as $1, ..., \lceil n/L \rceil$ where $L$ is the fixed batch size. Let the total variation of ground truth within batch $i$ be $C_i$. Time interval of batch $i$ is denoted by $[t_h^{(i)}, t_l^{(i)}]$. Due to bias variance decomposition within a batch we have,

$$R_i = \sum_{t=t_h^{(i)}}^{t_l^{(i)}} E[(x_t - \theta_t)^2] = (\theta_{t_h^{(i)}-1} - \theta_{t_h^{(i)}})^2 + \sum_{t=t_h^{(i)}+1}^{t_l^{(i)}} (\theta_t - \bar{\theta}_{t_h^{(i)}:t-1})^2 + \frac{\sigma^2}{t - t_h^{(i)}}, \quad (2)$$

$$\leq (\theta_{t_h^{(i)}-1} - \theta_{t_h^{(i)}})^2 + LC_i^2 + \sigma^2(2 + \log L),$$

with the convention $\theta_0 = 0$ and at start of bin our prediction is just the noisy realization of the previous data point.

Summing across all bins gives,

$$\sum_{i=1}^{\lceil n/L \rceil} R_i \leq LC_n^2 + 2\sigma^2 \frac{n(2 + \log L)}{L} + U^2 + C_n^2.$$

where we have used assumption (A4) to bound the bias of the first prediction. The above expression can be minimized by setting $L = \sqrt{n \log n} \frac{\sigma}{C_n}$ to yield a regret bound of $O(U^2 + C^2 + \sigma C_n \sqrt{n \log n})$ $\square$

**Theorem 12.** *Under the same setup as in Theorem 11, moving averages with window size $\frac{\sigma \sqrt{n}}{C_n}$ yields a dynamic regret of $O(\sigma C_n \sqrt{n} + U^2 + C_n^2)$*

*Proof.* Let the window size of moving averages be denoted by $m$. Consider the prediction at a time $x_t, t \geq m$. By bias variance decomposition we have,

$$E[(x_t - \theta_t)^2] = \left(\theta_i - \frac{\sum_{j=i-m}^{i-1} \theta_j}{m}\right)^2 + \frac{\sigma^2}{m}.$$

By Jensen's inequality,

$$\left(\theta_i - \frac{\sum_{j=i-m}^{i-1} \theta_j}{m}\right)^2 \leq \frac{\sum_{j=i-m}^{i-1}(\theta_j - \theta_i)^2}{m},$$

$$\leq \frac{2\sum_{j=i-m}^{i-1}(j - i + 1 + m)(\theta_{j+1} - \theta_j)^2}{m} \text{ ,by } (a + b)^2 \leq 2a^2 + 2b^2.$$

Notice that the term $(\theta_i - \theta_{i-1})^2$ will be multiplied by a factor $m$ in the above bias bound at time point $i$, $m - 1$ times in the next time point $i + 1$ and so on. By summing this bias bound across the times points, we obtain

$$\sum_{i=m}^{n} \frac{2\sum_{j=i-m}^{i-1}(j - i + 1 + m)(\theta_{j+1} - \theta_j)^2}{m} \leq 4m \sum_{i=1}^{n-1}(\theta_i - \theta_{i+1})^2 + U^2,$$

$$\leq 4mC_n^2 + U^2.$$

The squared bias for the initial points can be bounded by.

$$\sum_{i=1}^{m-1} (\theta_i - \hat{\theta}_{(1:i-1)})^2 \leq U^2 + C_n^2.$$

Summing the variance terms yields,

$$\sum_{t=1}^{n} \mathrm{Var}(x_t) = \sum_{t=1}^{m-1} \frac{\sigma^2}{t} + \sum_{t=m}^{n} \frac{\sigma^2}{m},$$

$$\leq \frac{(1 + \log m + n)\sigma^2}{m}.$$

Thus the total MSE can be minimized by setting $m = \frac{\sigma \sqrt{n}}{C_n}$, we obtain a dynamic regret bound of $O(\sigma C_n \sqrt{n} + U^2 + C_n^2)$

$\square$

## D    Proof of useful lemmas

We begin by recording an observation that follows directly from the policy.

**Lemma 13.** *For $m^{th}$ bin that spans the interval $[t_h^{(m)}, t_l^{(m)}]$, discovered by the policy, let the lengths of $\hat{\alpha}(t_h^{(m)} : t_l^{(m)} - 1)$ and $\hat{\alpha}(t_h^{(m)} : t_l^{(m)})$ be $k$ and $k^+$ respectively. Then $\sum_{l=0}^{\log_2(k)-1} 2^{l/2} \|\hat{\alpha}(t_h^{(m)} : t_l^{(m)} - 1)[l]\|_1 \leq \sigma$ and $\sum_{l=0}^{\log_2(k^+)-1} 2^{l/2} \|\hat{\alpha}(t_h^{(m)} : t_l^{(m)})[l]\|_1 > \sigma$*

Next we prove the marginal sub-gaussianity of the wavelet coefficients.

**Lemma 14.** *Consider the observation model $y_i = \theta_i + \sigma z_i$, where $z_i$ is iid sub-gaussian with parameter 1, $i = 1, .., n$. Let $\alpha_i$ denote the wavelet coefficients of the sequence $z = pad_0(y_1, ...y_n)$. Then each $\alpha_i$ is sub-gaussian with parameter $2\sigma$.*

*Proof.* Without loss of generaility let's charecterize $\alpha_1$. Let $\boldsymbol{u} = [u_1, ...u_n, u_{n+1}, \ldots, u_{|z|}]^T$ denote the first row of the orthonormal wavelet transform matrix. Then,

$$\alpha_1 = \sum_{i=1}^{n} y_i \left( u_i(1 - \frac{1}{n}) - \sum_{j=1, j \neq i}^{n} \frac{u_j}{n} \right).$$

Thus $\alpha_1$ is a differentiable function of iid sub-gaussian noise $z_i$. We can find its Lipschtiz constant by bounding the gradient w.r.t $z_i$ as follows,

$$\|\nabla \alpha_1(z_1, ..., z_n)\|_2 \leq \sigma \left( \sum_{i=1}^{n} 2u_i^2 (1 - \frac{1}{n})^2 + \frac{2}{n} \sum_{j=1, j \neq i}^{n} u_j^2 \right)^{\frac{1}{2}},$$

$$\leq \sigma (2 + 2)^{\frac{1}{2}},$$

$$= 2\sigma.$$

By proposition 2.12 in Johnstone [2017] we conclude that $\alpha_1$ sub-gaussian with parameter $2\sigma$.    $\square$

In the next lemma, we record the uniform shrinkage property of soft-thresholding estimator.

**Lemma 15.** *For any interval $[t_h, t_l]$, let $Y = pad_0(y_{t_h}, ..., y_{t_l})$ and $\Theta = pad_0(\theta_{t_h}, ..., \theta_{t_l})$. Then $|(T(HY))_i| \leq |(H\Theta)_i|$ with probability at-least $1 - 2n^{3-\beta/8}$ for each co-ordinate $i$.*

*Proof.* Consider a fixed bin $[\underline{l}, \overline{l}]$ with zero padded vector $Y \in R^k$. Due to sub-gaussian tail inequality, we have $|(HY)_i - (H\Theta)_i| \leq \sigma \sqrt{\beta \log(n)}$ with probability at-least $1 - 2/n^{\beta/8}$. Consider the case $(H\Theta)_i \geq \sigma \sqrt{\beta \log(n)}$. Then both the scenarios $(HY)_i \leq \sigma \sqrt{\beta \log(n)}$ and $(HY)_i > \sigma \sqrt{\beta \log(n)}$

leads to shrinkage to a value that is smaller than $|(H\Theta)_i|$ in magnitude due to soft-thresholding with threshold set to $\sigma\sqrt{\beta\log(n)}$. Now consider the case when $0 \leq (H\Theta)_i \leq \sigma\sqrt{\beta\log(n)}$. Again, soft-thresholding in both scenarios $(HY)_i \leq \sigma\sqrt{\beta\log(n)}$ and $\sigma\sqrt{\beta\log(n)} \leq (HY)_i \leq (H\Theta)_i + \sigma\sqrt{\beta\log(n)}$ leads to shrinkage to a value that is smaller than $|(H\Theta)_i|$ in magnitude. One can come up with a similar argument for the case where $(H\Theta)_i \leq 0$. Now applying a union bound across all $O(n)$ co-ordinates and all $O(n^2)$ bins, we get the statement of the lemma. $\qquad\square$

**Lemma 16.** *The number of bins, $M$, discovered by the policy is at-most* $\max\{1, 2n^{1/3}C_n^{2/3}\sigma^{-2/3}\log(n)\}$ *with probability at-least* $1 - 2n^{3-\beta/2}$.

*Proof.* Let $\Theta_m = [\theta_1^{(m)}, \theta_2^{(m)}, ..., \theta_p^{(m)}]^T$ be the mean subtracted and zero padded ground truth sequence values in $m^{th}$ bin $[\underline{l}, \bar{l}]$ discovered by our policy. $y^{(m)} = [y_1^{(m)}, y_2^{(m)}, ..., y_p^{(m)}]^T$ be the corresponding mean subtracted and zero padded observations. Note that due to zero padding $p \leq 2(\bar{l} - \underline{l})$ and some of the last values in the vector can be zeroes. Let $\alpha_m(\underline{l} : \bar{l}) = H\Theta$ denotes the discrete wavelet coefficient vector. We can view the computation of the Haar coefficients as a recursion. At each level $l$ of the recursion, the entire length $p$, is divided into $2^l$ intervals. Let the sample averages of elements of $\Theta_m$ in these intervals be denoted by the sequence $\{\tilde{\theta}_1, \tilde{\theta}_2, ..., \tilde{\theta}_{2^l}\}$. Let $\alpha_m^{(l)} \in \mathbb{R}^{2^l}$ denotes the vector of Haar coefficients at level $l$.

First note that the Haar coefficient $\alpha_m^{(l)}(i) = \frac{1}{2}\sqrt{\frac{p}{2^l}}(\tilde{\theta}_{2i} - \tilde{\theta}_{2i-1})$ with $i = 1, ..., 2^l$.

$$\|\alpha_m^{(l)}\|_1^2 \leq \frac{p}{2^{l+2}}\left(\sum_{i=1}^{2^l}|\tilde{\theta}_{2i} - \tilde{\theta}_{2i-1}|\right)^2,$$
$$\leq \frac{pTV^2[\underline{l}-1:\bar{l}]}{2^l},$$

where $TV[a,b]$ denotes the total variation of the true sequence in the interval $[a,b]$. The last inequality holds because the total variation of the smoothed sequence must be at-most four times the entire total variation of true sequence. The factor 4 is due to the fact that total variation when we pad a mean zero sequence with zeroes is at-most twice the total variation before zero padding.

We have,

$$\frac{1}{\sqrt{p}}\sum_{l=0}^{\log_2(p)-1}2^{l/2}\|\alpha_m^{(l)}\|_1 \leq \log p\, TV[\underline{l}-1:\bar{l}].$$

In the policy we compute $\hat{\alpha}_m(\underline{l} : \bar{l}) = T(Hy^{(m)})$ with the soft thresholding factor of $\sigma\sqrt{\beta\log(n)}$. From lemma 15, we have $|(T(Y))_i| \leq |(H\Theta)_i| \; \forall i \in [1,p]$ with probability at-least $1 - 2n^{3-\beta/8}$. Since $[\underline{l}, \bar{l}]$ is a bin discovered by policy, lemma 13 gives a lowerbound on $\|\alpha_m(\underline{l} : \bar{l})\|$. Putting it all together yields the relation,

$$\frac{\sigma}{\sqrt{p}} < \frac{1}{\sqrt{p}}\sum_{l=0}^{\log_2(p)-1}2^{l/2}\|\hat{\alpha}_m^{(l)}(\underline{l} : \bar{l})\|_1 \leq \frac{1}{\sqrt{p}}\sum_{l=0}^{\log_2(p)-1}2^{l/2}\|\alpha_m^{(l)}(\underline{l} : \bar{l})\|_1 \leq \log(p)\, TV[\underline{l}-1:\bar{l}], \quad (3)$$

with probability at-least $1 - 2n^{3-\beta/8}$.

Thus the total variation in the time interval $[\underline{l}-1, \bar{l}]$ can be lower bounded in probability as

$$TV[\underline{l}-1:\bar{l}] > \frac{\sigma}{\sqrt{p}\log n}.$$

Due to assumption $(A3)$ we have,

$$\sum_{i=1}^{M}TV[\underline{l}^i - 1 : \bar{l}^i] = C_n,$$

where $[\underline{l}^i : \bar{l}^i]$ are the bins discovered by the policy.

Let $p_i$ be the padded width of bin $i$ discovered by the policy. Then,

$$C_n \log n \geq \sum_{i=1}^{M} \frac{\sigma}{\sqrt{p_i}},$$

$$\geq \frac{M^2 \sigma}{\sum_{i=1}^{M} \sqrt{p_i}},$$

where the last line is obtained via Jensen's inequality. Now using Holder's inequality $\|x\|_\beta \leq d^{\frac{1}{\beta} - \frac{1}{\alpha}} \|x\|_\alpha$ for $0 < \beta \leq \alpha$, $x \in \mathbb{R}^d$ with $\alpha = 1/2$, $\beta = 1$ and noting that $\sum_{i=1}^{M} p_i \leq 2T$ gives,

$$\sigma M^2 \leq C_n \log n \sum_{i=1}^{M} \sqrt{p_i},$$

$$\leq C_n \log n \sqrt{Mn}.$$

Hence we get $M \leq (2n)^{1/3}(C_n \log n)^{2/3}\sigma^{-2/3} \leq 2n^{1/3}C_n^{2/3}\sigma^{-2/3}\log(n)$.

When $C_n = 0$, (3) implies that our policy will not restart with probability at-least $1 - 2n^{3-\beta/8}$ making $M = 1$. $\qquad\square$

We restate two useful results from Donoho [1995].

**Lemma 17.** *Consider the observation model $y = \alpha + Z$, where $y \in R^k$ and $|Z_i| \leq \delta \forall i \in [1, k]$. Let $\hat{\alpha}_\delta$ be the soft thresholding estimator with input $y$ and threshold $\delta$, then*

$$\|\hat{\alpha}_\delta - \alpha\|^2 \leq \sum_{i=1}^{k} min\{\alpha_i^2, 4\delta^2\}.$$

**Lemma 18.** *Consider the observation model $y = \alpha + Z$, where $y \in R^k$, $\alpha \in A$ and each $Z_i$ is sub-gaussian with parameter $\sigma^2$. If A is solid and orthosymmetric, then*

$$\inf_{\hat{\alpha}} \sup_{\alpha \in A} E[\|\hat{\alpha} - \alpha\|^2] \geq \frac{1}{2.22} \sup_A \sum_{i=1}^{k} min\{\alpha_i^2, \sigma^2\}.$$

Let's pause a moment to ponder how remarkable the above lemma is. The observations need not be even iid. Given $A$ is solid and orthosymmetric, all that is required is the marginal sub-gaussianity as the soft-thresholding operation works co-ordinate wise. Now we reprove theorem 4.2 from Donoho [1995] with a slight modification of threshold value in the estimator.

**Theorem 19.** *With probability at-least $1 - 2n^{-\beta/2}$, under the model in lemma 18, the soft thresholding estimator $\hat{\alpha}_\delta$ with $\delta = \sigma\sqrt{\beta \log(n)}$ obeys*

$$\|\hat{\alpha}_\delta - \alpha\|^2 \leq 8.88\beta(1 + \log(n)) \inf_{\hat{\alpha}} \sup_{\alpha \in A} E[\|\hat{\alpha} - \alpha\|^2]. \tag{4}$$

*Proof.* Consider the soft thresholding estimator $\hat{\alpha}_\delta$. By Gaussian tail inequality we have $P(\sup_i |Z_i| \geq \delta) \leq 2n^{-\beta/2}$. Conditioning on the event $\sup_i |Z_i| \leq \delta$ and applying lemma 17,

$$
\begin{aligned}
\|\hat{\alpha}_\delta - \alpha\|^2 &\leq \sum_{i=1}^{k} min\{\alpha_i^2, 4\delta^2\}, \\
&= \sum_{i=1}^{k} min\{\alpha_i^2, 4\beta\sigma^2 \log(n)\}, \\
&\leq max\{1, 4\beta \log(n)\} \sum_{i=1}^{k} min\{\alpha_i^2, \sigma^2\}, \\
&\leq (1 + 4\beta \log(n)) \sup_{\alpha \in A} \sum_{i=1}^{k} min\{\alpha_i^2, \sigma^2\}, \\
&\leq 4\beta(1 + \log(n)) \, 2.22 \inf_{\hat{\alpha}} \sup_{\alpha \in A} E[\|\hat{\alpha} - \alpha\|^2],
\end{aligned}
$$

where the last line follows from lemma 18. $\qquad\square$

It can be shown that wavelet coefficients of functions residing in the TV class is solid and orthosymmetric. As shown in lemma 14, the noisy wavelet coefficients are marginally sub-gaussian. Thus in the coefficient space, we are under the same observation model as in lemma 18. Using a uniform bound argument across all $O(n^2)$ bins and all $O(n)$ points within a bin along with lemma 14 leads to the following corollary.

**Corollary 20.** *The soft-thresholded wavelet coefficients of re-centered and zero padded noisy data within any interval $[t_h, t_l]$ satisfy relation* (4) *with probability atleast* $1 - 2n^{3-\beta/8}$.

Next, we record an important preliminary bound that will be used in proving the main result.

**Lemma 21.** *With probability at-least* $1 - \frac{\delta}{2}$, *the total squared error for online averaging between two arbitrarily chosen time points* $t_h$ *and* $t_l$ *satisfies*

$$
\sum_{t=t_h}^{t_l} (x_t^{t_h} - \theta_t)^2 \leq 4\sigma^2 \log(4n^3/\delta)(2 + \log(t_l - t_h + 1)) + 2(\theta_{t_h-1} - \theta_{t_h})^2 + 2\sum_{t=t_h+1}^{t_l} (\bar{\theta}_{t_h:t-1} - \theta_t)^2.
\tag{5}
$$

*Proof.* Throughout this lemma we assume the notation $\theta_0 = 0$. For proving this, first we bound the squared error for online sample averages within a bin, $b[\underline{l}, \bar{l}]$, that starts and ends at fixed times $\underline{l}$ and $\bar{l}$ respectively. Then a uniform bound argument will be used for bounding the squared error within any arbitrarily chosen bin. Note that $b[\underline{l}, \bar{l}]$ represents any fixed time interval and may not be even chosen by the policy. For $t \in [\underline{l}, \bar{l}]$, consider the prediction $x_t^l$, with same notation as used in the policy. Define a random variable $Z_t$ as

$$
Z_t = \frac{(x_t^l - \theta_t) - (\lambda_t - \theta_t)}{\sigma\sqrt{1/[t - \underline{l}]_{1+}}},
$$

where $[x]_{1+} = max\{1, x\}$, $\lambda_l = \theta_{l-1}$ and $\lambda_t = \bar{\theta}_{\underline{l}:t-1}, \forall t > \underline{l}$. $Z_t$ is subgaussian with variance parameter 1 and mean 0. Hence by sub-gaussian tail inequality, we have $P(|Z_t| \geq \sqrt{2\log(4/\delta)}) \leq \delta/2$. By noting that length of a bin is $O(n)$ and applying uniform bound across all time points within the current bin we have

$$
P\left(\sup_{\underline{l} \leq t \leq \bar{l}} |Z_t| \geq \sqrt{2\log(4n^3/\delta)}\right) \leq \delta/2n^2.
$$

Hence with probability at-least $1 - \delta/2n^2$,

$$
|x_t^l - \theta_t| \leq |\lambda_t - \theta_t| + \sigma\sqrt{\frac{2\log(4n^3/\delta)}{[t - \underline{l}]_{1+}}}, \forall t \in [\underline{l}, \bar{l}].
\tag{6}
$$

So the squared error within a bin can be bounded in probability as

$$\sum_{t=\underline{l}}^{\bar{l}}(x_t^l - \theta_t)^2 \le 2(\theta_{\underline{l}-1} - \theta_l)^2 + 2\sum_{t=\underline{l}+1}^{\bar{l}}(\bar{\theta}_{\underline{l}:t-1} - \theta_t)^2 + 2\sum_{t=\underline{l}}^{\bar{l}}\sigma^2 \frac{2\log(4n^3/\delta)}{[t-\underline{l}]_{1+}}.$$

Here we applied the inequality $(a+b)^2 \le 2a^2 + 2b^2$ on (6). Ultimately we are interested in analyzing the MSE within a bin detected by the policy. However the observations within a bin satisfies the restarting criterion of the policy and cannot be regarded independent. To break free of this constraint, we uniformly bound the quantity of interest — MSE here — across all possible bins. Noting that number of bins is $O(n^2)$ and applying uniform bound across all bins gives the following single sided tail bound.

Let E denote the event:
$\sup_{b[\underline{l}:\bar{l}]}(x_t^l - \theta_t)^2 - 2(\theta_{\underline{l}-1} - \theta_l)^2 - 2\sum_{t=\underline{l}+1}^{\bar{l}}(\bar{\theta}_{\underline{l}:t-1} - \theta_t)^2 - 2\sum_{t=\underline{l}}^{\bar{l}}\sigma^2 \frac{2\log(4n^3/\delta)}{[t-\underline{l}]_{1+}} \ge 0.$

Then,

$$P(E) \le \delta/2.$$

Hence with probability at-least $1 - \delta/2$, any bin $b[t_h : t_l]$ satisfies (5). □

Since (5) holds for any arbitrary interval of the time axis, it is particularly true for the bins discovered by the policy. Therefore the total squared error $T$ of the policy is upper bounded in probability by the sum of bin bounds of the form,

$$T \le \sum_{m=1}^{M} 4\sigma^2 \log(4n^3/\delta)(2 + \log(t_l^{(m)} - t_h^{(m)} + 1)) + 2(\theta_{t_h^{(m)}-1} - \theta_{t_h^{(m)}})^2 + 2\sum_{t=t_h^{(m)}+1}^{t_l^{(m)}}(\bar{\theta}_{t_h^{(m)}:t-1} - \theta_t)^2, \quad (7)$$

where the outer sum iterates over the bins and $M$ is the total number of bins. The first term inside the outer summation can be controlled if we can upper bound $M$. Now we set out to prove our main theorem.

# E   Proof of Theorem 1

From the discussion in section 1.1, the goal of bounding dynamic regret of the policy can be achieved by bounding the total squared error of its predictions. Our solution proceeds in two steps. First we upper bound the total squared error within a bin. Then we construct an upper bound for the number of bins spawned by the policy. With these two bounds in place, we bound the total squared error of the policy (7).

Let's first proceed to get a bound on the last summation term in (7). We use a reduction towards Follow The Leader (FTL) strategy. The term is basically the regret incurred by an FTL game with quadratic loss function for the duration $[t_h, t_l]$.

Let $\Theta(t_h : t_l - 1) = pad_0(\theta_{t_h}, ..., \theta_{t_l-1}) = [\Theta_{t_h}, ..., \Theta_{t_h+k-1}]^T$ denotes mean subtracted the zero padded true sequence in the interval $[t_h, t_l - 1]$. Then,

$$\sum_{t=t_h}^{t_l}(\bar{\theta}_{t_h:t-1} - \theta_t)^2 = (\bar{\theta}_{t_h:t_l-1} - \theta_{t_l})^2 + \sum_{t=t_h}^{t_l-1}(\bar{\theta}_{t_h:t-1} - \theta_t)^2,$$

$$\le (\bar{\theta}_{t_h:t_l-1} - \theta_{t_l})^2 + \sum_{t=t_h}^{t_l-1}\frac{(\bar{\theta}_{t_h:t-1} - \theta_t)^2}{(t - t_h + 1)} + \sum_{t=t_h}^{t_l-1}(\bar{\theta}_{t_h:t_l-1} - \theta_t)^2,$$

$$= (\bar{\theta}_{t_h:t_l-1} - \theta_{t_l})^2 + \sum_{t=t_h}^{t_l-1}\frac{(\bar{\theta}_{t_h:t-1} - \theta_t)^2}{(t - t_h + 1)} + \|\Theta(t_h : t_l - 1)\|^2. \quad (8)$$

We have applied FTL reduction for online game of predicting the true sequence $\theta_{t_h}, ..., \theta_{t_l-1}$ to get (8).

In the discussion below we assume that $\|D\theta_{1:n}\|_1 \le C_n$ and $|\theta_1| \le U$.

Now let's try to bound the term $\|\Theta(t_h : t_l - 1)\|_2^2$. This is basically the regret of the best expert. By triangle inequality,

$$
\|\Theta(t_h : t_l - 1)\|^2 \leq \|\hat{\alpha}(t_h : t_l - 1)\|_1^2 + \|\hat{\alpha}(t_h : t_l - 1) - \alpha(t_h : t_l - 1)\|_2^2,
$$

$$
\leq \left( \sum_{l=0}^{\log_2(p)-1} 2^{l/2} \|\hat{\alpha}(t_h : t_l - 1)[l]\|_1 \right)^2
$$

$$
+ \|\hat{\alpha}(t_h : t_l - 1) - \alpha(t_h : t_l - 1)\|_2^2, \tag{9}
$$

where $p$ is the padded length.

We can base our online averaging restart rule on the output of wavelet smoother. Suppose we decide to restart when $\|\hat{\alpha}(t_h : t_l)\|_1 \geq K n^{-1/3}$ for a constant $K$. Then the first term of (9) gives the optimal rate of $O(n^{1/3})$ when summed across all bins. But the estimation error term $\|\hat{\alpha}(t_h : t_l - 1) - \Theta(t_h : t_l - 1)\|^2$ should also be controlled. If the smoother is minimax over any bin $[t_h, t_l]$, then we can hope to get minimaxity over the entire horizon. However, the total variation inside the bin is not known to the smoother. This is where the adaptive minimaxity of wavelet smoother comes to rescue.

Suppose $\mathcal{F}$ denotes the class of functions $f$ with total variation $TV(f) \leq C_n$. Let $\mathcal{A}$ denote the set of all coefficients of the continuous wavelet transform of functions $f \in \mathcal{F}$. Then $\mathcal{A} \subset \Theta_{1,\infty}^{1/2}(C_n)$, where $\Theta_{1,\infty}^{1/2}(C_n)$ is a Besov body as defined in Donoho et al. [1998]. The minimax rate of estimation in this Besov body is $O(n^{-2/3} C_n^{2/3} \sigma^{4/3})$ where n is the number of observations. However, this is the rate of convergence of the $L_2$ function norm instead of the discrete (input-averaged) norm that we consider here. Over the Besov spaces, these two norms are close enough that the rates do not change (see section 15.5 of Johnstone [2017]). Hence Corollary 20 can be used to control the bias.

Let $\hat{y}(t_h : t)$ denotes the soft-thresholding estimates of the vector $pad_0(y_{t_h:t})$. i.e $\hat{y}(t_h : t) = H^T T(H \, pad_0(y(t_h : t)))$.

$$
(\bar{\theta}_{t_h:t_l-1} - \theta_{t_l})^2 \leq 2(\theta_{t_l-1} - \theta_{t_l})^2 + 2(\bar{\theta}_{t_h:t_l-1} - \theta_{t_l-1})^2,
$$

$$
\leq 2(\theta_{t_l-1} - \theta_{t_l})^2 + 4(\hat{y}(t_h : t_l - 1)[t_l - 1] - (\bar{\theta}_{t_h:t_l-1} - \theta_{t_l-1}))^2
$$

$$
+ 4(\hat{y}(t_h : t_l - 1)[t_l - 1])^2. \tag{10}
$$

Since L1 norm is greater than L2 norm, the policy's restart rule implies that

$$
(\hat{y}(t_h : t_l - 1)[t_l - 1])^2 \leq \sigma^2 \tag{11}
$$

Combining (10) and (11), we get

$$
(\bar{\theta}_{t_h:t_l-1} - \theta_{t_l})^2 \leq 2(\theta_{t_l} - \theta_{t_l-1})^2 + \gamma_1 (t_l - t_h)^{1/3} \, TV^{2/3}[t_h : t_l] \, \sigma^{4/3} + \sigma^2, \tag{12}
$$

where last line holds with probablity atleast $1 - 2n^{3-\beta/8}$ due to Corollary 20. Here $\gamma_1$ is a constant which can depend logarithmically on the width $t_l - t_h$.

Now let's bound the second term in (8). For any $t \in [t_h, t_l - 1]$ we have,

$$\sum_{t=t_h}^{t_l-1} \frac{(\bar{\theta}_{t_h:t-1} - \theta_t)^2}{(t - t_h + 1)} \le \sum_{t=t_h}^{t_l-1} \frac{2(\theta_t - \theta_{t-1})^2 + 2(\bar{\theta}_{t_h:t-1} - \theta_{t-1})^2}{t - t_h + 1},$$

$$\le \sum_{t=t_h}^{t_l-1} 2(\theta_t - \theta_{t-1})^2$$

$$+ \sum_{t=t_h}^{t_l-1} \frac{4(\hat{y}(t_h : t-1)[t-1] - (\bar{\theta}_{t_h:t-1} - \theta_{t-1}))^2 + 4(\hat{y}(t_h : t-1)[t-1])^2}{t - t_h + 1},$$

$$\le \sum_{t=t_h}^{t_l-1} 2(\theta_t - \theta_{t-1})^2 + (\gamma_2(t_l - t_h)^{1/3}\, TV^{2/3}[t_h : t_l]\, \sigma^{4/3} + 4\sigma^2)(1 + \log n),$$

(13)

where the last line holds with probability at-least $1 - 2n^{3-\beta/8}$.

$$\|\Theta(t_h : t_l - 1)\|_2^2 \le \left( \sum_{l=0}^{\log_2(p)-1} 2^{l/2}\|\hat{\alpha}(t_h : t_l - 1)[l]\|_1 \right)^2,$$

$$+ \gamma_3(t_l - t_h)^{1/3}\, TV^{2/3}[t_h : t_l]\, \sigma^{4/3},$$

$$\le \sigma^2 + \gamma_3(t_l - t_h)^{1/3}\, TV^{2/3}[t_h : t_l]\, \sigma^{4/3}, \tag{14}$$

with probability at-least $1 - 2n^{3-\beta/8}$ for some constant $\gamma_3$ which can depend logarithmically on the width $t_l - t_h$.

Due to Corollary 20 the bounds (12), (13), (14) all simultaneously holds with probability at-least $1 - 2n^{3-\beta/8}$. Combining these bounds, we get

$$\sum_{t=t_h}^{t_l} (\bar{\theta}_{t_h:t-1} - \theta_t)^2 \le 2\|D\theta_{t_h:t_l}\|_2^2 + \gamma(t_l - t_h)^{1/3}\, TV^{2/3}[t_h : t_l]\, \sigma^{4/3} + 6\sigma^2(1 + \log(n)),$$

with probability at-least $1 - 2n^{3-\beta/8}$ and $\gamma = \gamma_1 + \gamma_2(1 + \log(n)) + \gamma_3$.

When summed across all bins as in (7), with probability at-least $1 - 2n^{3-\beta/8}$ we have,

$$\sum_{m=1}^{M} \sum_{t=t_h^{(m)}}^{t_l^{(m)}} (\bar{\theta}_{t_h^{(m)}:t-1} - \theta_t)^2 \le U^2 + 2\|D\theta_{1:n}\|_2^2 + 6M\sigma^2(1 + \log n)$$

$$+ \sum_{m=1}^{M} \gamma\, (k^{(m)})^{1/3}\, TV^{2/3}[t_h^{(m)} : t_l^{(m)}]\, \sigma^{4/3},$$

$$\le U^2 + 2\|D\theta_{1:n}\|_2^2 + 6M\sigma^2(1 + \log n)$$

$$+ \gamma\sigma^{4/3}n^{1/3} \left( \sum_{m=1}^{M} \frac{k^{(m)}}{n} \right)^{\frac{1}{3}} \left( \sum_{m=1}^{M} TV[t_h^{(m)} : t_l^{(m)}] \right)^{\frac{2}{3}}, \tag{15}$$

$$\le U^2 + 2\|D\theta_{1:n}\|_2^2 + 6M\sigma^2(1 + \log n)$$

$$+ 2\gamma\sigma^{4/3}n^{1/3}C_n^{2/3}. \tag{16}$$

Here $k^{(m)}$ is the length of $\Theta(t_h^{(m)} : t_l^{(m)} - 1)$. The term $(\theta_{t_h^{(m)}-1} - \theta_{t_h^{(m)}})^2$ is at-most $U^2$ for the first bin. We arrive at (15) by applying Holder's inequality $x^T y \le \|x\|_p \|y\|_q$ with $p = 3$ and $q = 3/2$. For both (15) and (16) we use the fact that $\sum_{m=1}^M k^{(m)} \le 2n$ where the factor 2 is an artifact of zero-padding.

By appealing to lemma 16, we have with probability at-least $1 - 4n^{3-\beta/8}$,

$$\sum_{m=1}^M \sum_{t=t_h^{(m)}}^{t_l^{(m)}} (\bar{\theta}_{t_h^{(m)}:t-1} - \theta_t)^2 \le U^2 + 2\|D\theta_{1:n}\|_2^2 + 12\sigma^2 \log n$$
$$+ 24(\log(n))^2 n^{1/3} C_n^{2/3} \sigma^{4/3} + \gamma\sigma^{4/3} n^{1/3} C_n^{2/3}. \quad (17)$$

Next, we proceed to bound the first summation terms in (7). For this, we upperbound the number of bins to control the concentration terms in (7) when summed across all bins. Essentially our decision rule should not lead to over binning. Observe that the sum of total variations across all bins is $C_n$. If the decision rule guarantees (at-least in probability) that total variation inside any detected bin is $\tilde{\Omega}(n^{-1/3})$, then the number of bins is optimally $O(n^{1/3})$. Such a TV lower bounding property is satisfied by wavelet soft-thresholding as described in lemma 16. This is facilitated by the uniform shrinkage property of soft-thresholding estimator. More precisely,

Let's denote
$$V_m = 4\sigma^2 \log(2n^3/\delta)(2 + \log(t_l^{(m)} - t_h^{(m)} + 1)).$$

Then,

$$\sum_{m=1}^M V_m \le 4\sigma^2 \log(4n^3/\delta)(2 + \log(n)) \max\{1, 2n^{1/3} C_n^{2/3} \sigma^{-2/3} \log(n)\},$$
$$\le 4\sigma^2 \log(4n^3/\delta)(2 + \log(n))$$
$$+ 8n^{1/3} C_n^{2/3} \sigma^{4/3} \log(n) \log(4n^3/\delta)(2 + \log(n)), \quad (18)$$

with probability at-least $1 - 2n^{3-\beta/8}$. Here $[t_h^m, t_l^m]$ corresponds to the $m^{th}$ bin discovered by the policy. This relation follows due to Lemma 16.

Combining (18) and (17) we have with probability at-least $1 - 4n^{3-\beta/8} - \delta/2$

$$T \le 8n^{1/3} C_n^{2/3} \sigma^{4/3}(2 + \log(n)) \log(n)$$
$$+ 4\sigma^2 \log(4n^3/\delta)(2 + \log(n))$$
$$+ U^2 + 2\|D\theta_{1:n}\|_2^2 + 12\sigma^2 \log n$$
$$+ 24(\log(n))^2 n^{1/3} C_n^{2/3} \sigma^{4/3} + 2\gamma\sigma^{4/3} n^{1/3} C_n^{2/3}. \quad (19)$$

By observing that $\|D\theta_{1:n}\|_2 \le \|D\theta_{1:n}\|_1 = C_n$ we get the bound,

$$T \le 8n^{1/3} C_n^{2/3} \sigma^{4/3}(2 + \log(n)) \log(n)$$
$$+ 4\sigma^2 \log(4n^3/\delta)(2 + \log(n))$$
$$+ U^2 + 2C_n^2 + 12\sigma^2 \log n$$
$$+ 24(\log(n))^2 n^{1/3} C_n^{2/3} \sigma^{4/3} + 2\gamma\sigma^{4/3} n^{1/3} C_n^{2/3}.$$

The above bounds holds with probability at-least $1 - \delta$, if we set $\beta = 24 + \frac{8 \log(8/\delta)}{\log(n)}$.

We conclude our proof by observing that the above arguments can be readily extended to any batch smoother that satisfy the following criteria.

- Adaptive minimaxity over any interval within the time horizon.
- The restart decision rule optimally lowerbounds the total variation of any spawned bin.

Thus our policy can be viewed as a meta-algorithm that lifts a "well behaved smoother" to an optimal forecaster in the online setting.

Next we remark how the proof can be adapted to the setting where an extra boundedness constraint is put on the ground truth. i.e, $\theta_{1:n} \in TV(C_n)$ and $|\theta_i| \leq B, i = 1, \ldots, n$. Then the $U^2$ term in (19) becomes $B^2$. The additive $\|D\theta_{1:n}\|_2^2$ term can be bounded as,

$$
\begin{aligned}
\|D\theta_{1:n}\|_2^2 &= \sum_{i=2}^n (\theta_i - \theta_{i-1})^2, \\
&\leq \sum_{i=2}^n (|\theta_i| + |\theta_{i-1}|)(|\theta_i - \theta_{i-1}|), \\
&\leq 2BC_n.
\end{aligned}
$$

With the boundedness constraint, we also have $\|D\theta_{1:n}\|_2^2 \leq 4nB^2$. This essentially implies that $\|D\theta_{1:n}\|_2^2 \leq \min\{4nB^2, 2BC_n\}$.

Thus when $\|\theta_{1:n}\|_\infty \leq B$ and if we set $\beta = 24 + \frac{8\log(8/\delta)}{\log(n)}$ then with probability at-least $1 - \delta$,

$$
\begin{aligned}
T \leq\ & 8n^{1/3}C_n^{2/3}\sigma^{4/3}(2 + \log(n))\log(n) \\
& + 4\sigma^2\log(4n^3/\delta)(2 + \log(n)) \\
& + B^2 + 2\min\{4nB^2, 2BC_n\} + 12\sigma^2\log n \\
& + 24(\log(n))^2 n^{1/3}C_n^{2/3}\sigma^{4/3} + 2\gamma\sigma^{4/3}n^{1/3}C_n^{2/3}.
\end{aligned}
\tag{20}
$$

# F  Adaptive Optimality in Discrete Sobolev class

In this section, we establish that despite the fact that ARROWS is designed for the total variation class, it adapts to the optimal rates forecasting sequences that are more regular.

The discrete Sobelov class is defined as

$$
\mathcal{S}(C_n') = \{\theta_{1:n} : \|D\theta_{1:n}\|_2 \leq C_n'\}.
$$

The minimax cumulative error of nonparametric estimation in the discrete Sobolev class is $\theta_{1:n}(n^{2/3}[C_n']^{2/3}\sigma^{4/3})$ [see e.g., Sadhanala et al., 2016, Theorem 5 and 6].

Recall that the discrete Total Variation class that we considered in this paper is defined as

$$
\mathcal{T}(C_n) = \{\theta_{1:n} : \|D\theta_{1:n}\|_1 \leq C_n\}.
$$

By the norm inequalities, we know that

$$
\mathcal{T}(C_n') \subset \mathcal{S}(C_n') \subset \mathcal{T}(C_n'\sqrt{n}).
$$

The following refinement of our main theorem establishes that ARROWS also achieves the minimax rate in discrete Sobolev classes.

**Theorem 22.** *Let the feedback be $y_t = \theta_t + Z_t$ where $Z_t$ is an independent, $\sigma$-subgaussian random variable. Let $\theta_{1:n} \in \mathcal{S}(C_n')$. If $\beta = 24 + \frac{8\log(8/\delta)}{\log(n)}$, then with probability at least $1 - \delta$, ARROWS achieves a dynamic regret of $\tilde{O}(n^{2/3}[C_n']^{2/3}\sigma^{4/3} + U^2 + [C_n']^2 + \sigma^2)$ where $\tilde{O}$ hides a logarithmic factor in $n$ and $1/\delta$.*

*Proof.* Let's minimally expand the Sobolev ball to a TV ball of radius $C_n = \sqrt{n}C_n'$. This chosen radius of the TV ball is in accordance with the canonical scaling introduced in [Sadhanala et al., 2016]. This activates the following embedding:

$$
\mathcal{S}_1(C_n') \subseteq TV(C_n).
$$

We can rewrite (19) as

$$
\begin{aligned}
T \leq\; & 8n^{1/3}\|D\theta_{1:n}\|_1^{2/3}\sigma^{4/3}(2+\log(n))\log(n) \\
& + 4\sigma^2\log(4n^3/\delta)(2+\log(n)) \\
& + U^2 + 2\|D\theta_{1:n}\|_2^2 + 12\sigma^2\log n \\
& + 24(\log(n))^2 n^{1/3}\|D\theta_{1:n}\|_1^{2/3}\sigma^{4/3} + 2\gamma\sigma^{4/3}n^{1/3}\|D\theta_{1:n}\|_1^{2/3}.
\end{aligned}
\tag{21}
$$

The above representation reveals the optimality of our policy over Sobolev class $S_1(C_n')$. Enlarging the Sobolev class to the TV class that contains it does not change the minimax rate in the smoothing setting. See, e.g., Theorem 5 and 6 of [Sadhanala et al., 2016] and take $d = 1$, and $C_n' = n^{-1/2}C_n$. By using $\|x\|_1 \leq n^{1/2}\|x\|_2$ for $x \in \mathbb{R}^n$,

$$
\frac{\|D\theta_{1:n}\|_1}{n^{1/2}} \leq \|D\theta_{1:n}\|_2 \leq C_n' = \frac{C_n}{n^{1/2}}.
$$

Plugging this bound on $\|D\theta_{1:n}\|_1$ in (21) recovers the minimax regret for the Sobolev class of radius $C_n'$. The additional term of $\|D\theta_{1:n}\|_2^2$ — similar to as shown in in appendix I — is unavoidable in the online setting for predicting discrete Sobolev sequences.

$\square$

**Remark 23.** Note that $\mathcal{T}(C_n') \subset \mathcal{S}(C_n')$, therefore our lower bound from Proposition 6 still applies, which suggests that the additional $[C_n']^2 + \sigma^2$ is required and that ARROWS is an optimal forecaster for sequences in Sobolev classes as well.

## G    Fast Computation

We describe the proof of $O(n\log n)$ runtime guarantee below.

We use an inductive argument. Without loss of generality let the start of current bin be at time 1. Suppose we know the wavelet transform of points upto time $t$. Let the next highest power of 2 for both $t$ and $t+1$ be $p$. We identify this value as a pivot for time $t$ and $t+1$. Zero padding is done to hit this pivot. We can view the $pad_0$ operation at time $t+1$ as the difference between the padded original data and and a step signal. This step signal assume the value $\bar{y}_{1:t+1}$ in time $[1, t+1]$ and 0 in $[t+2, p]$. For computing wavelet transform of the step, we need to update only $O(\log(p))$ coefficients. Inputs to the Haar transform of the padded data at times $t$ and $t+1$ differs by just one co-ordinate. Hence coefficients of only $\log(p)$ wavelets need to be changed. Each such change can be performed in $O(1)$ time in an incremental fashion.

Now let's consider the case when the pivot for time $t+1$ is $2t$. Suppose we know the Haar wavelet coefficients upto time $t$. In this case, we need to compute the coefficients of $\log(t)$ newly introduced wavelets that span the interval $[t, 2t]$ since the zero padding will force most of the new wavelet coefficients to be zero. The computation of each of those new coefficients can be done in $O(1)$ due to sparsity of signal in interval $[t, 2t]$. We also need to change the first two wavelet coefficients which can be done again in in $O(1)$ time. In all these cases, we only need to do soft-thresholding to the newly updated coefficients. At the base case, when the pivot is just 2, then the computation can be in $O(1)$ time. Thus within a pivot $p$, the number of computations required is $O(p\log(p))$ which translates to $O(k^{(m)}\log(k^{(m)}))$ computations within the $m^{th}$ bin. Summing across all the bins yields a runtime complexity of $O(n\log(n))$.

## H    Regret of AOMD

In this section we prove that for any predictable sequence $\{M_t\}_{t=1}^n$, the AOMD algorithm has a dynamic regret of $\tilde{O}(\sqrt{n})$ when applied to our problem. As discussed in Section 2, consider loss functions $f_t(x) = (x - y_t)^2$ and comparator sequence $\{u_t\}_{t=1}^n$. First let's consider a deterministic noise setting [Donoho, 1995]:

$$
y_t = \theta_t + \delta\,\sigma\sqrt{20\log(n)},
$$

where $|\delta| \leq 1$ is chosen by a clever adversary. Let's proceed to get a bound on the quantity $D_n$. The gradient of our loss function is $2(x - y_t)$. So after observing the values of $x_t$ and $M_t$, an adversary can pick a suitable $\delta$ such that each term of $D_n$

$$D_n = \sum_{t=1}^{n} \|\nabla f_t(x_t) - M_t\|_*^2.$$

can be made $O(1)$. This gives an $O(n)$ bound for $D_n$.

We can show that $V_n$ is $O(n)$ if we assume that $\mathcal{X}$ is compact and all of the $y_t$ is bounded. Boundedness of $y_t$ follows from the assumptions (A3) and (A4). By appealing to assumption (A3) we see that

$$C_n(u_1, u_2, ..., u_n) = \sum_{t=1}^{n} \|u_t - u_{t-1}\|.$$

$C_n(\theta_1, ..., \theta_n)$ is $O(1)$. Plugging this into the regret bound specified in Jadbabaie et al. [2015] bounds the dynamic regret in our setting as $\tilde{O}(\sqrt{n})$.

We now relate this deterministic noise setting to the guassian setting where the observations are produced according to $y_t = \theta_t + Z_t$, where $Z_t$ is a zero mean sub gaussian with parameter $\sigma^2$. As described in proof of theorem 19, $P(\sup_i |Z_i| \geq \sigma \sqrt{20 \log(n)}) \leq 2n^{-9}$. Hence by conditioning on the event that $\sup_i |Z_i| \leq \sigma \sqrt{20 \log(n)}$, the regret bound of the deterministic noise setting applies to gaussian setting with high probability.

# I  Lower bound proof

*Proof of Proposition 6.* First, a lower bound of $\Omega(n^{1/3} C_n^{2/3} \sigma^{4/3})$ is given by [Donoho et al., 1998] for the smoothing estimator $x_{1:n}$ that has more information than we do. The argument uses the fact that the TV-ball is sandwiched between two Besov-bodies with identical minimax rate. To the best of our knowledge, the dependence on $C_n$ and $\sigma$ is first made explicit in, e.g., [Birge and Massart, 2001].

By the fact that "the max is larger than the mean", we have that for any prior distribution $\mathcal{P}$,

$$\sup_{\theta_{1:n} \in \mathrm{TV}(C_n)} \mathbb{E}\left[\sum_{t=1}^{n}(x_t - \theta_t)^2\right] \geq \mathbb{E}_{\theta_{1:n} \sim \mathcal{P}}\left[\mathbb{E}[\sum_{t=1}^{n}(x_t - \theta_t)^2 | \theta_{1:n}]\right].$$

Take $\mathcal{P}$ such that

1. $\theta_1 = U$ with probability 0.5 and $-U$ otherwise.

2. $\theta_2 = \theta_1 + C_n$ with probability 0.5 and $\theta_1 - C_n$ otherwise.

3. $\theta_t = \theta_2$ for $t = 3, 4, ..., n$.

Note that $x_1$ does not observe anything yet, therefore $x_1 = 0$ is the Bayes optimal decision rule. This gives a trivial lower bound of $\mathbb{E}\left[(x_1 - \theta_1)^2\right] \geq U^2$. Now, let's reveal $\theta_1$ to $x_2$ an additional information, then by the same argument, we have that $\mathbb{E}\left[(x_2 - \theta_2)^2\right] \geq C_n^2$.

Consider an alternative $\mathcal{P}$ when $\theta_1 = ... = \theta_n = \theta$. Let the noise be iid Gaussian with variance $\sigma^2$. In this case the problem reduces to a naive statistical estimation problem with $\theta \in [-U, U]$. For each $t$ which observes $t - 1$ iid samples from $\mathcal{N}(\theta, \sigma^2)$, then by Bickel et al. [1981], the minimax risk for this problem is

$$\inf_{\hat{\theta}} \sup_{\theta \in [-U, U]} \mathbb{E}(\hat{\theta} - \theta)^2 = \frac{\sigma^2}{t} - \frac{\pi^2 \sigma^4}{t U^2} + o(\frac{\sigma^4}{t U^2}).$$

Summing over $t = 2, 3, ..., n$, and apply the upper/lower bounds of the harmonic series, we have a lower bound of

$$\mathbb{E}\left[\sum_{t=1}^{n}(x_t - \theta_t)^2\right] \geq \max\{0, \sigma^2 \log(n + 1) - \frac{\pi^2 \sigma^4}{U^2}(1 + \log(n))(1 + o(1))\}.$$

Take the condition that $U > 2\pi\sigma$ and $n > 3$, the above expression can be further lower bounded by $0.5\sigma^2 \log(n)$. Note that this bound applies even if $C_n = 0$.

Finally, we can similarly apply the same argument to the case when $\theta_1 = 0$ and $\theta_2 = \ldots = \theta_n = \theta$ and where the constraint is that $-C_n \le \theta \le C_n$. This gives us a lower bound of

$$\mathbb{E}\left[\sum_{t=2}^{n}(x_t - \theta_t)^2\right] \ge \max\{0, \sigma^2 \log(n) - \frac{\pi^2\sigma^4}{C_n^2}(1 + \log(n-1))(1 + o(1))\}.$$

If $C_n > 2\pi\sigma$ and $n > 3$, we can again bound it below by $0.5\sigma^2 \log(n)$. In other word, we get the $\sigma^2 \log(n)$ lower bound provided that either $C_n$ or $U$ is greater than $2\pi\sigma$.

The proof is complete by taking the average of lower bounds above. We can take $c = 1/6$.   $\square$

## I.1   Lower bound with extra boundedness constraint on ground truth

Suppose we assume $|\theta_i| \le B, i = 1, \ldots, n$. Then we can adapt the proof presented above by considering a prior $\mathcal{P}$ such that $\theta_i = \epsilon_i B, i = 1, \ldots, \min\{n, 1 + \lfloor C_n/2B\rfloor\}$. $\theta_i = \theta_{1+\lfloor C_n/2B\rfloor}, \forall i > \min\{n, 1+\lfloor C_n/2B\rfloor\}$. Here $\epsilon_i$ are independent random variables assuming value $+1$ with probability $0.5$ and $-1$ with probability $0.5$. Assume that we reveal to learner the probability law of observations $\theta_i$. Under this setting we can see that $\mathbb{E}\left[\sum_{t=1}^{n}(x_t - \theta_t)^2\right] \ge B^2 + \min\{(n-1)B^2, BC_n/2\}$.

Under the setting of $y_i = \theta_i + \epsilon_i$ for iid $\sigma^2$ sub-gaussian $\epsilon_i$, $|\theta_i| \le B$ and $i = 1, \ldots, n$,[Donoho et al., 1990] establishes that minimax total squared error scales as $n\min\{B^2, \sigma^2\}$. This along with previous discussions imply that in the bounded ground truth setting the minimax risk is $\tilde{\Omega}\left(\min\{nB^2, n\sigma^2, n^{1/3}C_n^{2/3}\sigma^{4/3}\} + B^2 + \min\{nB^2, BC_n\} + \sigma^2\right)$.

## I.2   Minimax regret using ARROWS for bounded ground truth

From (20) the regret of ARROWS $T_{\text{ARROWS}}$ satisfy

$$T_{\text{ARROWS}} = \tilde{O}(n^{1/3}C_n^{2/3}\sigma^{4/3} + \min\{nB^2, BC_n\} + \sigma^2).$$

Let $T_1$ be the regret of an algorithm, say $\mathcal{A}_1$, that predicts $p \sim N(0, \sigma^2)$ at time step 1 and zero for remaining times. Then it can seen that

$$T_1 = O(nB^2 + \sigma^2),$$
$$= O(nB^2 + \sigma^2 + \min\{nB^2, BC_n\}).$$

Let $T_2$ be the regret of an algorithm, say $\mathcal{A}_2$, that predicts $y_{t-1}$ at time $t$. Then,

$$T_2 = O(n\sigma^2 + \min\{nB^2, BC_n\}).$$

Now consider running exponentially weighted average forecaster [Cesa-Bianchi and Lugosi, 2006] with three experts: ARROWS, $\mathcal{A}_1$ and $\mathcal{A}_2$. Since squared error is exponentially concave, by Proposition 3.1 of [Cesa-Bianchi and Lugosi, 2006] such a forecaster when run with $\eta = 2$ gives a regret $T$ that satisfy,

$$T = O\left(\min\{T_{\text{ARROWS}}, T_1, T_2\} + \log 3\right),$$
$$= \tilde{O}\left(\min\{nB^2, n\sigma^2, n^{1/3}C_n^{2/3}\sigma^{4/3}\} + B^2 + \min\{nB^2, BC_n\} + \sigma^2 + \log 3\right).$$

Thus we acheive the optimal cumulative squared error upto a small additive term of $\log 3$. If we look at the per round regret this additive term contributes to a small $O(1/n)$ quantity.

## I.3   Connections to other lower bounds in literature

[Besbes et al., 2015] derived a lower bound of $O(n^{1/2}V_n^{1/2})$ by packing a sequence of quadratic loss functions. Note that this is larger than the upper bound that we attain with quadratic losses.

Though this observation seems confusing, a careful study reveals that there is no contradiction. For constructing the lowerbound, [Besbes et al., 2015] used a variational budget $V_n$ as , $V_n = \sum_{t=2}^{n} \sup_{x \in conv(\theta_1, \dots \theta_n)} |f_t(x) - f_{t-1}(x)| = \sum_{t=2}^{n} \sup_{x \in [\theta_{min}, \theta_{max}]} |(x - \theta_t)^2 - (x - \theta_{t-1})^2|$, where $conv(.)$ denotes the convex hull of a sequence of points. This is different from the variational budget they use in section 2 of their paper and is also different from $C_n$ that we use for the TV class. When applied to our setting this $V_n$ is no longer proportional to our $C_n$, instead, it is proportional to $(\theta_{max} - \theta_{min}) C_n$.

The packing set constructed through the functions defined in equation (A-12) of [Besbes et al., 2015] obeys $(\theta_{max} - \theta_{min}) = \frac{1}{2} V_n^{1/4} n^{-1/4}$. So we have $C_n = \frac{V_n}{V_n^{1/4} n^{-1/4}} = V_n^{3/4} n^{1/4}$, where we have subsumed proportionality constants. Thus we see that $V_n = \frac{C_n^{4/3}}{n^{1/3}}$. Putting this into their lowerbound recovers exactly our $n^{1/3} C^{2/3}$ bound.

The additional $C_n^2$ term that appears in our upper bound is required for any methods that do online forecasting of sequences in the TV class. The reason why OGD appears to not require $C_n^2$ according to [Besbes et al., 2015] is because they require the $\theta_t$ to be bounded for all $t$, while we only require $\theta_1$ to be bounded by $U$ (see Theorem 11).

The lowerbound discussed in [Yang et al., 2016] considers a more general setting of smooth non-strongly convex sequence of loss functions. Such a lowerbound will not apply in our more restrictive setting.

## J   Optimality of linear forecasters in Discrete Sobolev class

In this section we first establish that just like ARROWS, linear strategies such as OGD and MA are also optimal forecasters for sequences in Discrete Sobolev class. Then we substantiate it using experiments.

**Theorem 24.** *Let the feedback be $y_t = \theta_t + Z_t$ where $Z_t$ is an independent, $\sigma$-subgaussian random variable. Let $\theta_{1:n} \in \mathcal{S}(C_n')$. Restarting OGD with batch size of $\frac{\sigma^{2/3}(n \log n)^{1/3}}{[C_n']^{2/3}}$ achieves an expected dynamic regret of $\tilde{O}(U^2 + [C_n']^2 + n^{2/3}[C_n']^{2/3}\sigma^{4/3})$.*

*Proof.* We stick to the same notations as in Appendix C. Let's start the analysis from (2). Let $t' = t - t_h^{(i)}$.

$$(\theta_t - \bar{\theta}_{t_h^{(i)}:t-1})^2 \leq \frac{\left(\sum_{i=t_h^{(i)}}^{t-1}(\theta_t - \theta_i)\right)^2}{[t']^2},$$

$$\leq \frac{t'}{[t']^2} \sum_{i=t_h^{(i)}}^{t-1}(\theta_t - \theta_i)^2,$$

$$\lesssim L[C_i']^2.$$

Hence summing across all points yields,

$$R_i \lesssim L^2[C_i']^2 + \sigma^2 \log L.$$

So the total expected regret becomes,

$$\sum_{i=1}^{\lceil n/L \rceil} R_i \lesssim L^2[C_n']^2 + \frac{n}{L}\sigma^2 \log L.$$

By choosing $L = \frac{\sigma^{2/3}(n \log n)^{1/3}}{[C_n']^{2/3}}$ we get the theorem. The additive term $[C_n']^2$ arises similarly as in proof of Theorem 11 □

The optimality of Moving Averages can be proved similarly.

**Remark 25.** Thus from Theorems 3, 9, 11, 24 we see that ARROWS is minimax over both the classes $TV(C_n)$ and $\mathcal{S}(C_n/\sqrt{n})$ while linear forecasters such as OGD and MA require different tuning parameters to perform optimally in each class.

Next, we give numerical experiments substantiating the claims.

**Experimental results:** Here we consider a doppler function $f(t) = \sin\left(\frac{2\pi(1+\epsilon)}{t/n+0.01}\right)$ with $n$

Figure 6: Regret plot for policies calibrated according to Sobolev radius for a Doppler function

being the time horizon. For this function $C'_n = \|D\theta\|_2 = O(C_n/\sqrt{n})$ when $n$ is sufficiently large and $\|D\theta\|_2 = O(C_n)$ for small $n$ for a TV bound $C_n = O(1)$. Thus for sufficiently large $n$, this sequence belong to a small Sobolev ball with radius $O(1/\sqrt{n})$ while the TV class that encloses that Sobolev ball as per Theorem 22 has radius $O(1)$.

We observe noisy data $y_i = f(i/n) + z_i, i = 1, ..., n$ and $z_i$ are iid normal variables with $\sigma = 1$. Figure 6 plots the regret averaged across 5 runs in a log log scale. The necessary input calibration was made as per Remark 23 while running ARROWS. We can see that in this case all the algorithms perform in an optimal manner.

Specifically we identify two regimes one for small $n$ and other for larger $n$. When $n$ is large, we obtain the minimax regret rate $\tilde{O}(n^{1/3})$ due to small $C'_n$ which can be considered as $O(1/\sqrt{n})$. Numerically for $n > 10^5$, $C'_n$ is less than 0.1% of $C_n$. For smaller values of $n$ where $C'_n$ can be not too small, we attain a regret in accordance with the $\tilde{O}(n^{2/3})$ minimax rate. Numerically when $n < 10^4$, $C'_n$ is atleast 8.5% of $C_n$ which can be considered as $O(C_n) = O(1)$.