[Reviews · NeurIPS 2019]

Reviewer 1



=== Update after reading the authors' response: The authors' response answered my questions well. One limitation that I missed in my initial review is that the coordinates of the parameter theta (the true signal sequence) are not assumed to be bounded; the only assumption is that theta lies in a total variation ball. This means that the only bound on these coordinates is through the bound C on the total variation of the sequence. Hence, the dependence on the bound B on the sequence (L-infty norm of theta) is implicit and replaced by the worst-case upper bound C, which leads to a dependence on C^2 instead of B*C on the intersection of those balls. I think that this limitation should be addressed, given that in the online learning literature it is more customary to provide the explicit dependence on B. To be specific, this would entail assuming that theta lies in the intersection of a TV ball of radius C and an L-infinity ball of radius B (where both B and C can be assumed to be known, though adaptivity to those can be considered), and providing matching upper and lower bound over this class. This appears to be feasible; otherwise, the authors should at least explicitly mention this point. I chose to keep my initial score, given the merits of this paper. However, I encourage the authors to address or at least mention the above issue in the final version of the paper. == This paper considers the problem of sequential prediction of a sequence of real values. The sequence is assumed to be obtained by adding independent noise to a true sequence; the quality of predictions is measured by the cumulative squared error between the true and predicted values. This quantity essentially coincides with the dynamic regret from the online learning literature. Since one cannot hope for a sublinear regret without any assumption on the sequence, the authors consider sequences that are bounded in total variation. Total variation balls are often considered as rich nonparametric classes in the nonparametric estimation literature, which unlike the smaller Holder and Sobolev balls can account for abrupt changes and inhomogeneous smoothness. These classes motivated successful developments such as wavelet shrinkage in the 1990s. Achieving the optimal rate over these classes is not trivial, and requires some spatially adaptive algorithms; in particular, linear smoothers, a large class of algorithms which includes kernel smoothers, are know to fare suboptimally over these classes. The problem considered by the authors is actually different from the denoising problem considered in the nonparametric literature (where the goal is to estimate the true past signal in a batch fashion); here, the goal is to forecast the next observation online. The main contributions are: - Given the knowledge of the time horizon n, the total variation C and the noise level sigma, the authors propose an efficient algorithm called ARROWS with optimal n^{1/3} (C \sigma^2)^{2/3} regret. The fact that the O(n^{1/3}) regret can be achieved can also be seen from the general results on online nonparametric regression of (among others) Rakhlin and Sridharan, although this latter approach is non-constructive. - Through a connection with the problem of denoising, the authors note that linear algorithms such as online gradient descent (OGD) can only achieve suboptimal O(n^{1/2}) regret over such classes. - Finally, the authors show that for some range of values of C, the problem of forecasting is actually harder than that of denoising. I found this paper to be very interesting and well-written. The main points were articulated clearly, and I appreciated the connection between the two bodies of work. (I did not read the proofs in the supplementary material.) Remarks: * Theorem 8, and remark (lines 257-258) "despite the fact that ARROWS is designed for total variation class, it adapts to the optimal rates of forecasting sequences that are spatially regular": doesn't this apply to any minimax strategy for the TV class (by the same argument)? In this case, it would be more precise to formulate this (in this section and the introduction) as a consequence of the optimality on TV classes, rather than another property specific to the ARROWS algorithm. * Often in online learning, the considered notion of regret is obtained by constraining the comparator class, but not the sequence of values (which is seen as deterministic and arbitrary, although bounded). Do your results carry to this formulation? Typos: l. 102 "then" -> "than"; l. 225 "bound the number"

Reviewer 2



This paper is generally well-written and quite clear. The contributions seem significant though the setting may be a bit narrow with the squared loss only, no exogenous variables, known variance, horizon and bound on the total variation... I would be interested in more details on real applications (or experiments with real dataset rather same simulations) and with how this setting may be extended to the use of covariates. Other comments: - In the classical notion of dynamic regret (see e.g., Zinkevich 03), the complexity is measured in terms of the total variation of the comparison sequence (x_t), not the one of the true sequence (theta_t). This has the advantage to provide some kind of oracle inequality. It would be nice to discuss the optimal bound in that case and if the current analysis might be extended to it. - If I understood well, the improvement over Besbe et al. 2015, is due to the squared loss (instead of general strongly convex losses) which allows a bias-variance separation in the analysis. Is this only possible for squared loss or might this be true for a wide class of loss functions (such as self-concordant losses)? - I did not understand very well how this setting falls into the framework of online non-parametric regression. I would enjoy more explanations. Furthermore, it is mentioned in the introduction (l.148) that the algorithm provided here is the first polynomial time algorithm with O(n^(1/3)) regret though in the Appendix it is said that the one of Gaillard and Gerchinovitz 2015 has a complexity O(n^(7/3)). Besides, could your lower-bound (with the C^2 term) be re-obtained from their minimax rate? - Would the term C^2 in the lower-bound be necessary if we assumed all the theta to be bounded? - How the algorithm could be adapted to unknown meta-parameters: horizon (n), noise variance (sigma) and bound on the total variation? I am wondering if a doubling trick or a meta-aggregation using an expert advice algorithm would be sufficient. By the way, sigma should be added as a meta-parameter of the algorithm.

Reviewer 3



The algorithm is shown to have regret of the optimal order n^{1/3} with probability 1-\delta. The algorithm runs in O(n\log n) time. The infimum in the definition of regret (the displayed formula on page 2) seems misplaced.

[Author Response · NeurIPS 2019]

We thank all the reviewers for insightful comments and suggestions.

**Reviewer 1:** Thanks for the spot-on comments and for being our champion! We address the two remarks below.

Remark 1: In the batch setting, optimality in TV class indeed implies optimality in enclosed Sobolev and Holder classes
as the reviewer pointed out. However it is not true for forecasting due to the dependence of $[C'_n]^2$ in the optimal regret
rate as in theorem 8. While bounding the regret of ARROWS, we get a ground truth dependent L2 norm term $\|D\theta\|_2^2$ in
equation (20). This enables the adaptive minimaxity for Sobolev and Holder classes. However, a minimax strategy
whose regret bound contains the term $\|D\theta\|_1^2$ in the place of $\|D\theta\|_2^2$ in (20) will be optimal for TV class but fails to get
the correct dependence on $[C'_n]^2$ for the Sobolev class.

Remark 2: Achieving minimax forecasting in the TV-constrained comparator setting with a polynomial time algorithm is
an intriguing open question. Our results do not directly apply to that stronger setting. Although some of our techniques
might be reusable but we believe nontrivial new algorithmic ideas/proof techniques are probably needed. Our work is
better viewed under the lens of non-stationary sequential stochastic optimization as in Besbes et al [1] with squared
error loss and noisy gradient feedback.

**Reviewer 2:** Thanks for the detailed and insightful review. Please see Remark 2 above on the comparison to the
TV-constrained comparator setting and detailed responses to other questions.

*Re More general loss functions:* Generalization to other convex costs is regarded as a future work. Thanks for the
suggestion of self-concordant losses. It is a good direction to explore.

*Re Relation to Gaillard and Gerchinovitz[2015]:* The regret bound of $O(n^{1/3})$ in [2] attained by an $O(n^{7/3})$ runtime
policy holds for Holder class which features more regular functions than TV class. Their regret bound in theorem 11
fails to capture the optimal dependence on the Lipschitz constant and hence cannot be used to construct the correct
lowerbound with precise dependence on all of the problem parameters in our setting.

*Re Boundedness of theta and $C_n^2$ term in the lowerbound:* If we assume all theta to be bounded by $B$ then we would
be able to get a better $\Omega(BC_n)$ bound. For instance we can consider packing functions that alternates $C_n/B$ times
between $0$ and $B$. This also points to the fact that forecasting is harder than smoothing. However, this boundedness
constraint implies that we will be focusing only on a smaller subset of all sequences whose TV is bounded by $C_n$. Of
course this $B$ in worst case is at most $U + C_n$ where $U$ is the bound on first data point.

*Re Adaptivity to $C_n$:* Adaptivity to unknown variational functionals are usually nontrivial. Contrary to the reviewer's
comment, the uniform restarting proposed in [1] is in fact not adaptive. It requires knowing $C_n$ to set the optimal
restarting intervals. To the best of our knowledge, Zhang et al. 2018 [3] was the first paper that made it adaptive —
albeit suboptimally in our setting — as $\sqrt{C_n}$ to the total variation. Even there, they achieve adaptivity with a very nice
new idea of connecting to strongly adaptive regret minimizing algorithms.

That said, the reviewer's question challenged us to look into the problem further. We are now convinced that with
a simple tweak in the restart rule, it is possible to transform ARROWS to **an anytime algorithm that optimally**
**adapts to** $C_n$ — the TV of ground truth. Let the expression in LHS of the restart rule be $\hat{C}$. The idea is to
replace $n$ and $C_n$ in the RHS of restart rule by $k$ and $\hat{C}$ respectively. So we restart when $\hat{C} > \sigma k^{-1/2}$. All
the results can be proved to be true with this almost fully adaptive restart scheme. We do not have space for a
proof in this short rebuttal, instead we present below but the regret plot with the new restart rule as an empirical
validation. We will include this update in main paper if accepted. $\sigma$ if unknown, can be robustly estimated (thanks
to sparsity of the wavelet coeffs of Bounded Variation functions) using first few observations as mentioned in line 69.
40

**Reviewer 3:** Thanks for appreciating our contributions!

**References:**

[1] Besbes et al. *Non-stationary stochastic optimization*, In Operations
Research 2015.

[2] Gaillard et al. *A chaining algorithm for online nonparametric regres-*
*sion.* In COLT 2015.

[3] Zhang et al. *Dynamic Regret of Strongly Adaptive Methods*, In ICML
2018.

Figure 1: *Regret plot for function in Fig.2 of main paper with the new restart scheme that makes ARROWS optimally adaptive to both $n$ and $C_n$.*

[Meta-Review · NeurIPS 2019]

The paper introduces an optimal online learning algorithm for the non-parametric class of total-variation bounded sequences. Moreover, it shows that a class of well-known existing methods cannot be optimal, and that there is a separation between the achievable rates for estimation and prediction in terms of regret. These are all fundamental results, and the paper clearly discusses their connections to important developments in non-parametric statistical estimation. Comments for the final version: Please carefully consider the reviews, and take the (updated) comments into account when preparing the final version of the paper. In particular, please consider the dependence on B mentioned by reviewer #1. Note that there is already a well-known online learning algorithm called AROW (Crammer, Kulesza, Dredze, NeurIPS 2009), so please change the algorithm name from ARROWS to something else.